# Bridging the Divide: Reconsidering Softmax and Linear Attention

**Dongchen Han**[∗]  **Yifan Pu**[∗]  **Zhuofan Xia**[∗]  **Yizeng Han**  **Xuran Pan**

**Xiu Li**  **Jiwen Lu**  **Shiji Song**  **Gao Huang**[†]

Tsinghua University

## Abstract

Widely adopted in modern Vision Transformer designs, Softmax attention can effectively capture long-range visual information; however, it incurs excessive computational cost when dealing with high-resolution inputs. In contrast, linear attention naturally enjoys linear complexity and has great potential to scale up to higher-resolution images. Nonetheless, the unsatisfactory performance of linear attention greatly limits its practical application in various scenarios. In this paper, we take a step forward to close the gap between the linear and Softmax attention with novel theoretical analyses, which demystify the core factors behind the performance deviations. Specifically, we present two key perspectives to understand and alleviate the limitations of linear attention: the injective property and the local modeling ability. Firstly, we prove that linear attention is not injective, which is prone to assign identical attention weights to different query vectors, thus adding to severe semantic confusion since different queries correspond to the same outputs. Secondly, we confirm that effective local modeling is essential for the success of Softmax attention, in which linear attention falls short. The aforementioned two fundamental differences significantly contribute to the disparities between these two attention paradigms, which is demonstrated by our substantial empirical validation in the paper. In addition, more experiment results indicate that linear attention, as long as endowed with these two properties, can outperform Softmax attention across various tasks while maintaining lower computation complexity. Code is available at https://github.com/LeapLabTHU/InLine.

## 1 Introduction

Recent years have witnessed the unprecedented success of Transformer and attention [32] in the field of computer vision [7]. Softmax attention, also known as dot-product attention, has demonstrated remarkable expressive power, leading to state-of-the-art performance across various vision tasks [30, 2, 23, 35]. However, applying Softmax attention in vision also faces challenges. The quadratic complexity of Softmax attention results in prohibitively high computational cost when applied with a global receptive field. Previous works [33, 19, 36, 6, 45] have strived to reduce the computation complexity by restricting receptive fields or introducing sparsity. Although effective, these approaches inevitably compromise Softmax attention's ability for long-range modeling and scalability.

The nature of Softmax attention forces to compute the dot-products between queries and keys $QK^{\top} \in \mathbb{R}^{N \times N}$ at first, and then aggregates values $V \in \mathbb{R}^{N \times C}$ by the normalized score $\mathrm{Softmax}(QK^{\top}/\sqrt{d})$, which accounts for the quadratic $\mathcal{O}(N^2)$ complexity *w.r.t.* the sequence length $N$. On the contrary, linear attention relaxes the similarity score between $Q$ and $K$ from Softmax to other functions

---

[∗]Equal Contribution.

[†]Corresponding Author.

38th Conference on Neural Information Processing Systems (NeurIPS 2024).

which can be decomposed into kernels, *i.e.*, the linear attention replaces the original score function $\text{Sim}(Q, K) = \text{Softmax}(QK^\top/\sqrt{d})$ with $\text{Sim}(Q, K) = \phi(Q)\phi(K)^\top$, where $\phi(\cdot)$ is the kernel function. This substitution enables a change in the computation order from $\left(\phi(Q)\phi(K)^\top\right)V$ to $\phi(Q)\left(\phi(K)^\top V\right)$ based on the associative law of matrix multiplication, reducing the complexity from $\mathcal{O}(N^2)$ to $\mathcal{O}(N)$ *w.r.t.* the sequence length $N$. Nevertheless, every coin has two sides. Linear attention proves to be less effective than Softmax attention [3, 27, 40], whose poor expressive power limits its practical application. Although many pieces of research [27, 28, 22, 9, 40] have attempted to alleviate this issue in different ways, we still do not have a complete understanding of the key factors that contribute to the gap between linear and Softmax attention.

In this paper, we delve into the fundamental differences between linear and Softmax attention, offering two insightful perspectives to demystify the topic: the injective property and local modeling capability. Firstly, we consider attention as a function that maps a query to an attention score. We find that the injectivity of this attention function greatly affects the performance of the model. Specifically, if the attention function is not injective, different queries will induce identical attention distributions, leading to severe semantic confusion within the feature space. Our rigorous analysis has demonstrated that the Softmax attention function is an injective function, whereas the linear attention function is not. Therefore, linear attention is vulnerable to the semantic confusion problem, which largely leads to its insufficient expressiveness. Secondly, our analysis of the attention weight distribution has confirmed that the success of Softmax attention is not solely dependent on its strong long-range modeling capabilities. Effective local modeling is also crucial to achieving optimal outcomes.

To validate our analyses, we present two simple yet effective methods to endow linear attention with the injective property and the local modeling ability, respectively. The widely employed Swin Transformer [19] architecture is used to validate our findings. The results highlight the importance of both properties in the gap between linear and Softmax attention. Moreover, comprehensive experiments demonstrate that linear attention, endowed with these two properties, outperforms the widely used Softmax attention across diverse tasks.

Our main contributions and takeaways are summarized as follows: **(1) Injectivity** is a key disparity between linear and Softmax attention. While Softmax attention is injective, the non-injective nature of linear attention causes semantic confusion and severely impairs model performance. To the best of our knowledge, our work is the first to conceptualize attention as a mapping function and prove the vital importance of its injective property. **(2) Local modeling** is still essential to the effectiveness of the attention mechanism, even though it is renowned for its large receptive field and outstanding long-range modeling ability. **(3)** We challenge the viewpoint that linear attention is inferior to Softmax attention and demonstrate that with the above two properties, linear attention can outperform Softmax attention while maintaining lower computation complexity.

## 2 Related Works

**Vision Transformer and Softmax Attention.** Vision Transformer [7] is the pioneer work that introduces self-attention to vision. Since then, attention has found success in various vision tasks [7, 2, 24]. The widely used attention mechanism is Softmax attention [32], also known as dot-product attention, which computes the similarity between all query-key pairs. Although effective, its quadratic computation complexity leads to unmanageable cost when processing global feature maps. Therefore, various approaches [19, 34, 36, 14, 45, 12] have been proposed to reduce the computational overhead of Softmax attention. PVT [33] employs downsampling of keys and values to reduce computational complexity. Swin Transformer [19] restricts the receptive field by introducing window attention pattern. NAT [14] mimics convolution and calculates attention within the neighborhood of each feature, and DAT [36] presents an input-dependent sparse attention pattern.

**Linear Attention.** As opposed to Softmax attention, Linear attention is another attention paradigm with a natural linear complexity of $\mathcal{O}(N)$. Linear attention replaces Softmax with kernel functions, thereby reducing computational complexity to linear through a change in computation order. Nonetheless, prior studies [27, 3, 29, 40, 11] have demonstrated that linear attention performs markedly worse than Softmax attention. CosFormer attributed this discrepancy to the efficient re-weighting mechanism of Softmax attention and proposed cosine re-weighting to enhance linear attention. Nyströmformer [38] and SOFT [22] use matrix decomposition to further approximate Softmax operation. Efficient Attention [28] applies Softmax function to queries and keys. TransNormer [26] identifies

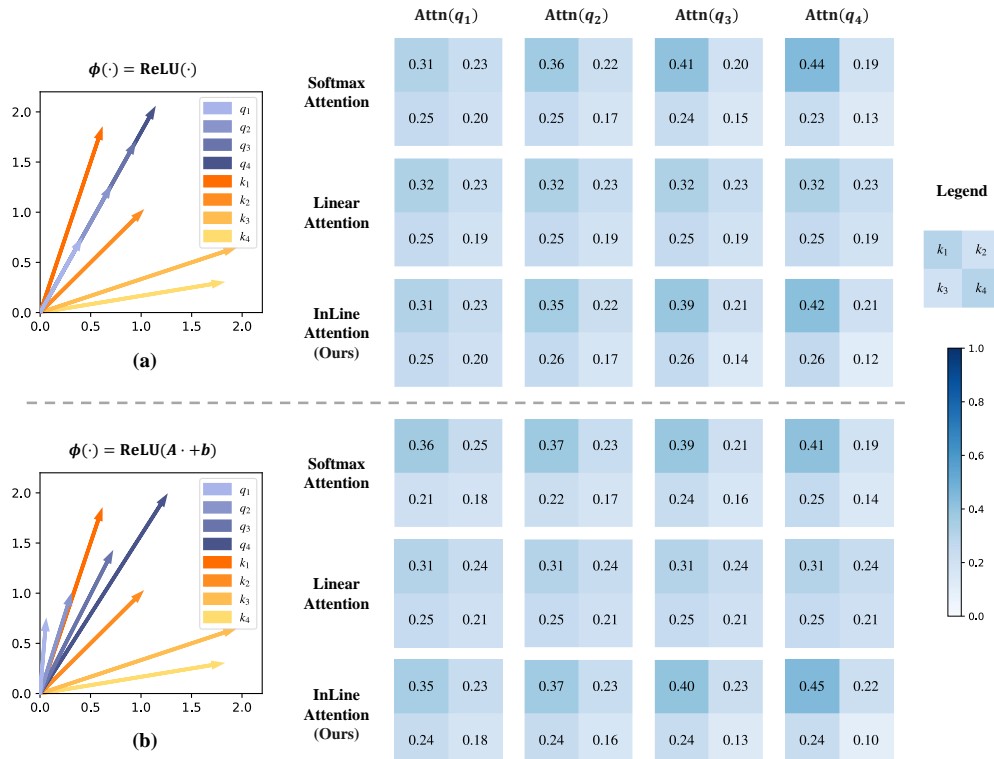

Figure 1: **An illustration of injective property and confusion problem**. Non-injectivity leads to various semantic confusions in linear attention when different kernel functions are employed. (a) With $\phi(\cdot) = \text{ReLU}(\cdot)$, linear attention assigns the same attention values to collinear queries of varying lengths. (b) Using $\phi(\cdot) = \text{ReLU}(A \cdot + b)$, linear attention faces severe confusion problem, producing identical attention distribution for certain queries with different directions and lengths.

that unbounded gradients and attention dilution harm linear attention. FLatten Transformer [9] introduces a focused function to address the over-smoothing issue. MLLA [10] draws inspiration from Mamba [8] to improve linear attention.

Despite their elegant outcomes, the fundamental reason for the disparity between linear attention and Softmax attention remains unclear. In this work, we perform an in-depth analysis of the disparities between linear and Softmax attention, identifying two crucial properties of high-performance Softmax attention: injectivity and local modeling ability. We present both theoretical proofs and experimental verification to validate our findings.

## 3 Preliminaries

**Attention Formulation**. Let $x \in \mathbb{R}^{N \times C}$ be an input of $N$ tokens. In each self-attention head, $x$ is transformed into $Q = xW_Q, K = xW_K, V = xW_V$ through projection matrices $W_{Q/K/V} \in \mathbb{R}^{C \times d}$, where $C$ and $d$ are the channel dimension of module and each head. Therefore, we have $Q, K, V \in \mathbb{R}^{N \times d}$, and $Q_i, K_i, V_i \in \mathbb{R}^d$. Based on this, Softmax attention [32] computes the attention weights and calculates the output as the weighted sum of value:

$$S_i = \left[ \frac{\exp(Q_i^\top K_1)}{\sum_{j=1}^N \exp(Q_i^\top K_j)}, \cdots, \frac{\exp(Q_i^\top K_N)}{\sum_{j=1}^N \exp(Q_i^\top K_j)} \right]^\top, \quad O_i^S = S_i^\top V. \quad (1)$$

For simplicity, we omit $\sqrt{d}$ in $\exp(Q_i^\top K_j / \sqrt{d})$ since we can equivalently renormalize $Q$ and $K$. This attention paradigm has been highly successful in modern vision Transformers. However, it should compute the similarity between all query-key pairs, resulting in $\mathcal{O}(N^2)$ complexity. Consequently, employing Softmax attention with a global receptive field results in overwhelming computation cost.

Linear attention [15] was proposed to efficiently handle the computation challenge with linear complexity of $\mathcal{O}(N)$. Specifically, $\exp(Q_i^\top K_j)$ is replaced by $\phi(Q_i)^\top \phi(K_j)$, where $\phi$ is kernel function. In this way, linear attentions reformulate eq. (1) as:

$$L_i = \left[ \frac{\phi(Q_i)^\top \phi(K_1)}{\sum_{j=1}^N \phi(Q_i)^\top \phi(K_j)}, \cdots, \frac{\phi(Q_i)^\top \phi(K_N)}{\sum_{j=1}^N \phi(Q_i)^\top \phi(K_j)} \right]^\top,$$

$$O_i^L = L_i^\top V = \sum_{j=1}^N \frac{\phi(Q_i)^\top \phi(K_j)}{\sum_{j=1}^N \phi(Q_i)^\top \phi(K_j)} V_j^\top = \frac{\phi(Q_i)^\top (\sum_{j=1}^N \phi(K_j) V_j^\top)}{\phi(Q_i)^\top (\sum_{j=1}^N \phi(K_j))}. \tag{2}$$

The form of $O_i^L$ suggests that explicitly computing attention weights $L_i$ is unnecessary. Instead, we can change the computation order from $(\phi(Q)\phi(K)^\top)V$ to $\phi(Q)(\phi(K)^\top V)$ based on the associative property of matrix multiplication. By doing so, the computation complexity is reduced to $\mathcal{O}(N)$.

**Injective Property**. Let $f : A \to B$ be a mapping function. We call $f$ an injective function if and only if $\forall x, y \in A, x \neq y$, it holds that $f(x) \neq f(y)$.

# 4 Analysing the Gap between Linear and Softmax Attention

Due to its linear computation complexity, linear attention is considered a promising solution to address the computational challenges of Softmax attention in high-resolution scenarios. However, previous works [27, 3, 40] have shown that linear attention's expressive power is significantly lower than that of Softmax attention, rendering it impractical for real-world applications. In this section, we conducted an in-depth analysis of the gap between linear and softmax attention from two perspectives: injective mapping and local modeling capability, and offer both theoretical proofs and experimental verification to enhance understanding of the key disparities between these two attention types.

## 4.1 Injectivity of Attention Function

We first define the function of Softmax and linear attention as follows:

$$S_K, L_K : \mathbb{R}^d \to \mathbb{R}^N, \quad S_K(Q_i) = S_i, \quad L_K(Q_i) = L_i, \tag{3}$$

where $Q_i$ denotes the query, and $S_i, L_i$ are the attention scores in eq. (1) and eq. (2). Given keys $K \in \mathbb{R}^{N \times d}$, $S_K, L_K$ can be viewed as the function of query $q$, mapping each $q$ to its corresponding Softmax and linear attention scores, $S_K(q)$ and $L_K(q)$. Then the final outputs of Softmax and linear attention corresponding to $q$ can be formulated as $O^S = S_K(q)^\top V$ and $O^L = L_K(q)^\top V$.

**Injective property**. In this work, we identify that the *injective property* of the attention function significantly impacts model performance, which may largely contribute to the gap between linear and Softmax attention. Specifically, we prove that under mild assumptions, the Softmax attention function $S_K$ is injective, whereas linear attention function $L_K$ is not (Proposition 1 and 2. Please refer to Appendix for complete proof). As a consequent, for two different queries $p$ and $q$ ($p \neq q$), Softmax attention should produce different attention distributions $S_K(p) \neq S_K(q)$, while linear attention may yield the same linear attention values $L_K(p) = L_K(q)$. Since different queries $p \neq q$ typically represent distinct semantics, the non-injective property of linear attention actually leads to semantic confusion, i.e. $L_K(p) = L_K(q)$ and $O_p^L = L_K(p)^\top V = L_K(q)^\top V = O_q^L$, making the model unable to distinguish certain semantics.

**Proposition 1** (Softmax attention is injective) *Given* $K \in \mathbb{R}^{N \times d}$ *with* $\mathrm{rank}(K) = d$ *and* $\mathrm{rank}([K, \mathbf{1}_{N \times 1}]) = d + 1$. $\forall p, q \in \mathbb{R}^d, p \neq q$, *we have* $S_K(p) \neq S_K(q)$.

**Proposition 2** (Linear attention is not injective) *Let* $\phi : \mathbb{R}^d \to \mathbb{R}^d$ *be a continuous function.* $\exists p, q \in \mathbb{R}^d, p \neq q$, s.t. $L_K(p) = L_K(q)$.

We provide an example to better understand the injective property and confusion problem. As shown in Fig. 1(a), there are four collinear vectors with different lengths. Benefiting from injectivity, Softmax attention ensures that each of these four queries obtains distinct attention scores, producing more focused attention distributions for longer queries. Nevertheless, with kernel function $\phi(\cdot) = \mathrm{ReLU}(\cdot)$, linear attention fails to distinguish the same semantics with different intensities, i.e. collinear queries with varying lengths, resulting in identical attention scores for all these four queries. Consequently,

linear attention is unable to yield more focused attention scores for stronger semantics, which may explain the lack of focus ability discussed in [9]. When using kernel functions with stronger nonlinearity, linear attention encounters more pronounced confusion issues. For instance, in Fig. 1(b), employing kernel function $\phi(\cdot) = \mathrm{ReLU}(A \cdot + b)$, linear attention assigns exactly the same attention scores to four queries with different directions and lengths. This serious semantic confusion can directly impair model's performance.

**Confusion problem in real models**. While Fig. 1 illustrates the concept of confusion, it is also crucial to verify if this issue occurs in real models. Therefore, we conduct statistical analysis based on Deit-T. We count the occurrences of confusion for each image (i.e., $p \neq q$ but $\mathrm{S_K}(p) = \mathrm{S_K}(q)$ or $\mathrm{L_K}(p) = \mathrm{L_K}(q)$) during inference on the ImageNet [5] validation set. As it is rare for two vectors to be strictly equal in floating-point representation, we consider them approximately equal if the L2 norm of their difference is less than 1e-3. The results are provided in Fig. 2. Almost all samples did not encounter confusion on model employing Softmax attention, whereas a large number of samples encountered confusion more than $2^5$ times on linear attention model. This proves the existence of confusion problem with linear attention in real models.

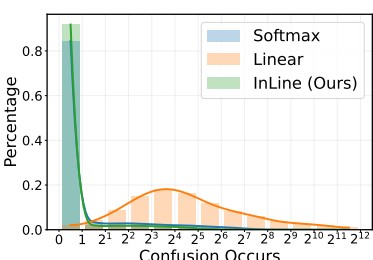

Figure 2: The distribution of the number of times each image encounters confusion during inference.

**The importance of injectivity**. We further verify the importance of injectivity by inducing confusion in Softmax attention. To achieve this, we apply additional non-injective mapping functions to each query before the Softmax attention calculation, i.e., introducing $Q_i = f(Q_i)$ prior to eq. (1), where $f$ is a non-injective function. Specifically, we use $f_1(q) = \frac{\mathrm{ReLU}(q)}{\|\mathrm{ReLU}(q)\|}$ to simulate the confusion observed in linear attention using the kernel function $\phi(\cdot) = \mathrm{ReLU}(\cdot)$, as depicted in Fig. 1(a), and employ $f_2(q) = \frac{\mathrm{ReLU}(Aq+b)}{\|\mathrm{ReLU}(Aq+b)\|}$ to replicate the confusion in Fig. 1(b). As shown in Tab. 1, introducing confusion leads to an obvious decrease in performance, underscoring the crucial role of the attention function's injective property. Therefore, the non-injectivity of linear attention is likely a key factor leading to its limited expressive capacity.

Table 1: Introducing confusion to Softmax attention.

| Confusion | None | $f_1(q) = \frac{\mathrm{ReLU}(q)}{\|\mathrm{ReLU}(q)\|}$ | $f_2(q) = \frac{\mathrm{ReLU}(Aq+b)}{\|\mathrm{ReLU}(Aq+b)\|}$ |
|---|---|---|---|
| Acc. | 72.2 | 70.6 | 69.9 |

**Make linear attention injective**. We propose a simple yet effective solution to make linear attention an injective function. The proof of Proposition 2 (see Appendix) demonstrates that $\forall \alpha \neq 0, \alpha\phi(p)$ obtains identical scores in linear attention due to the omission of $\alpha$ in division, resulting in non-injectivity. Hence, we simply transform the normalization of linear attention from division to subtraction, presenting our **injective linear attention (InLine)** as follows:

$$\mathrm{InL_K}(Q_i) = \left[\phi(Q_i)^\top \phi(K_1), \cdots, \phi(Q_i)^\top \phi(K_N)\right]^\top - \frac{1}{N}\sum_{s=1}^{N}\phi(Q_i)^\top\phi(K_s) + \frac{1}{N}, \qquad (4)$$

and the attention output corresponding to $Q_i$ can be written as $O_i^I = \mathrm{InL_K}(Q_i)^\top V$. This modification ensures that the attention weights still sum up to 1, while transforming the attention function into an injective one (see Proposition 3). Thus, injective linear attention can distinguish different queries, akin to Softmax attention, and it no longer suffers from confusion problem.

**Proposition 3** (InLine attention is injective) *Let $\phi : \mathbb{R}^d \to \mathbb{R}^d$ be an injective map. Given $K \in \mathbb{R}^{N \times d}$ with $\mathrm{rank}(\phi(K)) = d$ and $\mathrm{rank}([\phi(K), \mathbf{1}_{N \times 1}]) = d+1$. $\forall p, q \in \mathbb{R}^d, p \neq q, \Rightarrow \mathrm{InL_K}(p) \neq \mathrm{InL_K}(q)$.*

Additionally, similar to linear attention, InLine attention can be calculated with $\mathcal{O}(N)$ complexity by changing the computation order:

$$O_i^I = \mathrm{InL_K}(Q_i)^\top V = \sum_{j=1}^{N}\left[\phi(Q_i)^\top\phi(K_j) - \frac{1}{N}\sum_{s=1}^{N}\phi(Q_i)^\top\phi(K_s) + \frac{1}{N}\right]V_j^\top$$

$$= \phi(Q_i)^\top\left[\sum_{j=1}^{N}\phi(K_j)V_j^\top\right] - \left[\phi(Q_i)^\top\sum_{j=1}^{N}\phi(K_j) - 1\right]\frac{1}{N}\sum_{j=1}^{N}V_j. \qquad (5)$$

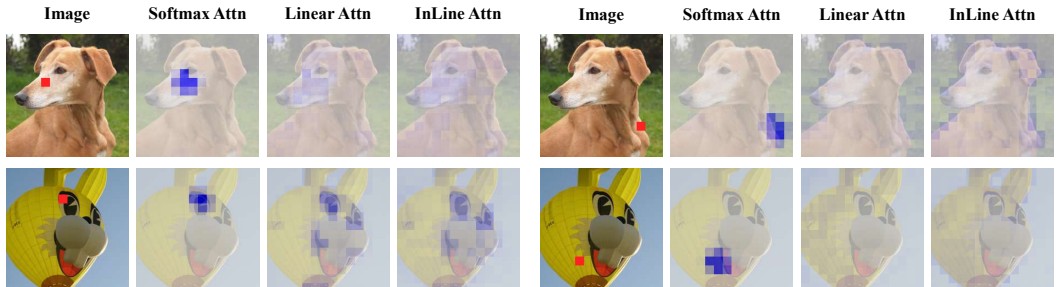

| Image | Softmax Attn | Linear Attn | InLine Attn | Image | Softmax Attn | Linear Attn | InLine Attn |

Figure 4: Visualizations of attention distributions. Softmax attention exhibits strong local bias. The other two attention types yield meaningful attention distributions, but focus more on global modeling.

Table 2: Model performances on ImageNet-1K when masking out tokens from different positions. Loc. $k \times k$ means masking out tokens in local $k \times k$ windows for each query. Rand $n$ represents randomly masking out $n$ tokens out of local $3 \times 3$ windows for each query. The attention scores of each query still sum up to $1$. These models are tested directly without retraining.

| Mask Out Position | None | Loc. $3 \times 3$ | Loc. $5 \times 5$ | Loc. $7 \times 7$ | Rand 9 | Rand 25 | Rand 49 |
|---|---|---|---|---|---|---|---|
| Softmax Attn | 72.2 | 51.6 | 24.3 | 9.0 | 71.7 | 71.5 | 71.1 |
| InLine Attn | 70.0 | 58.0 | 40.0 | 20.0 | 70.0 | 69.9 | 69.5 |

Since we can compute $\sum_{j=1}^{N} \phi(K_j)$ and $\frac{1}{N}\sum_{j=1}^{N} V_j$ once and reuse them for every query, the overall complexity of InLine attention is $2Nd^2 + Nd = \mathcal{O}(Nd^2)$. Accounting for multi-head, the complexity becomes $\mathcal{O}(NCd)$, where $C$ and $d$ are the channel dimension of the model and each head.

## 4.2 Local Modeling Capability

Attention mechanism is famous for its large receptive field and outstanding long-range modeling capability. However, we find that effective local modeling is crucial for the effectiveness of attention.

In Fig. 3, we compute the sum of attention values assigned to local $3 \times 3$ neighborhoods for each query using DeiT-T. With a total of $14 \times 14 + 1 = 197$ tokens in each attention layer of DeiT-T, if attention scores are randomly assigned, the expected sum of attention for a $3 \times 3$ neighborhood would be $\frac{9}{197}$. The result shows that all three attention paradigms tend to pay more attention to the neighborhoods of each query, revealing local bias, especially in shallow layers. Notably, Softmax attention allocates a substantial amount of attention to local windows, suggesting a stronger local modeling ability compared to the other two attention paradigms. Visualizations are provided in Fig. 4 to further confirm this finding.

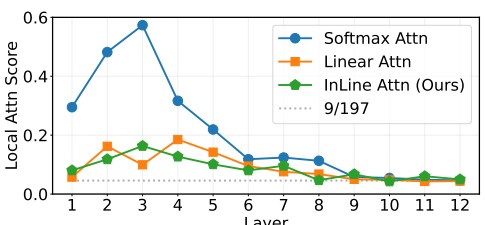

Figure 3: The sum of attention scores in the local 3×3 windows of each query from DeiT-T.

We speculate that Softmax attention's superior performance stems from robust local priors and strong local modeling capabilities. To validate this hypothesis, we employ attention masks to mask out tokens from various positions and assess their effect on model performance. The results are presented in Tab. 2. Two key observations emerge: 1. Masking out local tokens significantly decreases model performance, while randomly masking out the same number of tokens has a minor impact on results. 2. Softmax attention's performance suffers more severely than InLine attention when local tokens are masked out. These findings demonstrate the significance of local modeling for both attention types and prove that Softmax attention's advantage over InLine attention primarily attributes to its stronger local modeling ability.

Based on our analysis, increasing local bias may enhance the expressive power of InLine attention. In light of this, we employ a MLP to predict additional local attention residual for InLine attention.

Table 3: Ablation on the impact of injective property using Swin-T.

| Kernel Function $\phi(\cdot)$ | ReLU$(\cdot)$ | ReLU$(A \cdot + b)$ | LeakyReLU$(\cdot)$ | Identity$(\cdot)$ |
|---|---|---|---|---|
| Linear Attn | 77.3 | 70.2 | 1.5 | 0.2 |
| InLine Attn | 79.8 | 80.0 | 79.8 | 80.2 |

Table 4: Ablation on local modeling ability based on Swin-T. Identity$(\cdot)$ kernel function is used.

| | Window | FLOPs | #Param | Acc. | | Window | FLOPs | #Param | Acc. |
|---|---|---|---|---|---|---|---|---|---|
| InLine-Swin-T | $7^2$ | 4.5G | 30M | 80.3 | InLine-Swin-T | $7^2$ | 4.5G | 30M | 81.6 |
| w/o res. | $14^2$ | 4.5G | 30M | 80.4 | w/ res. | $14^2$ | 4.5G | 30M | 82.1 |
| | $28^2$ | 4.5G | 30M | 80.2 | | $28^2$ | 4.5G | 30M | 82.3 |
| | $56^2$ | 4.5G | 30M | 80.2 | | $56^2$ | 4.5G | 30M | 82.4 |

Specifically, the output corresponding to $Q_i$ is defined as:

$$O_i = \text{InL}_K(Q_i)^\top V + \sum_{j=1}^{9} r_j V_j^{N(i)}, \;\; r = \text{MLP}(\overline{x}), \tag{6}$$

where $\text{InL}_K(\cdot)$ denotes InLine attention function, $\overline{x}$ is the average of input tokens, $r$ is the predicted local attention residual, and $V_j^{N(i)}$ represents the value in the $3 \times 3$ neighborhood of $Q_i$. In this way, we explicitly enhance InLine attention's local bias by introducing local attention residual term. We refer to InLine attention with local attention residual, i.e. eq. (6), as **InLine attention module**. As the local residual term introduces little computational cost $Nd + d^2 + 9Nd$, the InLine attention module still maintains a linear complexity of $\mathcal{O}(N)$.

## 5 Empirical Study

In Sec. 4, we analyzed two core factors behind the performance gap between Softmax and linear attention, proposing possible remedies. In this section, we conduct empirical verification to fully validate the importance of these two properties and the effectiveness of our methods.

### 5.1 Implementation

We utilize the popular Swin Transformer architecture [19] to investigate the effects of injectivity and local modeling capability. Specifically, we substitute the original Softmax attention in Swin-T with linear attention to establish the baseline model. Subsequently, we introduce the injective property and local bias in turn to assess their respective impacts. To fully verify the effectiveness of InLine attention module, we further apply it to four advanced and representative Transformer models including DeiT [30], PVT [33], Swin [19], CSwin [6] and offer broad comparisons with various state-of-the-art methods using Softmax attention.

### 5.2 Datasets and Experiment Details

**ImageNet classification**. The ImageNet-1K [5] recognition dataset contains 1.28M training images and 50K validation images with a total of 1,000 classes. For a fair comparison, we train our model using identical settings as the corresponding baseline model. We use AdamW [21] optimizer to train all our models from scratch for 300 epochs, employing cosine learning rate decay with 20 epochs of linear warm-up. The initial learning rate is $1 \times 10^{-3}$, and the weight decay is 0.05. Augmentation and regularization strategies consist of RandAugment [4], Mixup [42], CutMix [41], and random erasing [43]. Following CSwin [6], EMA [25] is used in the training of InLine-CSwin models.

**COCO object detection**. COCO [18] object detection and instance segmentation dataset has 118K training and 5K validation images. We follow the training and testing strategies of the corresponding baseline model and employ pretrained InLine backbones to conduct experiments.

**ADE20K semantic segmentation**. ADE20K [44] is a well-established benchmark for semantic segmentation which encompasses 20K training images, 2K validation images and 150 semantic categories. The same setting as baseline model is adopted.

Table 5: Comparison with baseline models on ImageNet-1K. See full comparison table in Appendix.

| Method | Reso | #Params | FLOPs | Top-1 | Method | Reso | #Params | FLOPs | Top-1 |
|---|---|---|---|---|---|---|---|---|---|
| DeiT-T [30] | $224^2$ | 5.7M | 1.2G | 72.2 | Swin-T [19] | $224^2$ | 29M | 4.5G | 81.3 |
| **InLine-DeiT-T** | $224^2$ | 6.5M | 1.1G | **74.5** (+2.3) | **InLine-Swin-T** | $224^2$ | 30M | 4.5G | **82.4** (+1.1) |
| DeiT-B | $224^2$ | 86.6M | 17.6G | 81.8 | Swin-S | $224^2$ | 50M | 8.7G | 83.0 |
| **InLine-DeiT-B** | $448^2$ | 23.8M | 17.2G | **82.3** (+0.5) | **InLine-Swin-S** | $224^2$ | 50M | 8.7G | **83.6** (+0.6) |
| PVT-S | $224^2$ | 24.5M | 3.8G | 79.8 | Swin-B | $224^2$ | 88M | 15.4G | 83.5 |
| **InLine-PVT-S** | $224^2$ | 21.6M | 3.9G | **82.0** (+2.2) | **InLine-Swin-B** | $224^2$ | 88M | 15.4G | **84.1** (+0.6) |
| PVT-L | $224^2$ | 61.4M | 9.8G | 81.7 | Swin-B | $384^2$ | 88M | 47.0G | 84.5 |
| **InLine-PVT-L** | $224^2$ | 50.2M | 10.2G | **83.6** (+1.9) | **InLine-Swin-B** | $384^2$ | 88M | 45.2G | **85.0** (+0.5) |

Table 6: Comparison with SOTA methods on ImageNet-1K.

| Method | Reso | #Params | FLOPs | Top-1 | Method | Reso | #Params | FLOPs | Top-1 |
|---|---|---|---|---|---|---|---|---|---|
| PVTv2-B2 [34] | $224^2$ | 25M | 4.0G | 82.0 | Swin-B [19] | $224^2$ | 88M | 15.4G | 83.5 |
| ConvNeXt-T [20] | $224^2$ | 29M | 4.5G | 82.1 | PVTv2-B5 [34] | $224^2$ | 82M | 11.8G | 83.8 |
| Focal-T [39] | $224^2$ | 29M | 4.9G | 82.2 | ConvNeXt-B [20] | $224^2$ | 89M | 15.4G | 83.8 |
| MViTv2-T [17] | $224^2$ | 24M | 4.7G | 82.3 | Focal-B [39] | $224^2$ | 90M | 16.4G | 84.0 |
| CSwin-T [6] | $224^2$ | 23M | 4.3G | 82.7 | CSwin-B | $224^2$ | 78M | 15.0G | 84.2 |
| DiNAT-T [13] | $224^2$ | 28M | 4.3G | 82.7 | NAT-B [14] | $224^2$ | 90M | 13.7G | 84.3 |
| InLine-CSwin-T | $224^2$ | 21M | 4.3G | 83.2 | InLine-CSwin-B | $224^2$ | 73M | 14.9G | 84.5 |
| ConvNeXt-S [20] | $224^2$ | 50M | 8.7G | 83.1 | Swin-B [19] | $384^2$ | 88M | 47.0G | 84.5 |
| PVTv2-B3 [34] | $224^2$ | 45M | 7.9G | 83.2 | CaiT-S36 [31] | $384^2$ | 68M | 48.0G | 85.0 |
| CSwin-S [6] | $224^2$ | 35M | 6.9G | 83.6 | ConvNeXt-B [20] | $384^2$ | 89M | 45.0G | 85.1 |
| Focal-T [39] | $224^2$ | 51M | 9.4G | 83.6 | MViTv2-B [17] | $384^2$ | 52M | 36.7G | 85.2 |
| MViTv2-S [17] | $224^2$ | 35M | 7.0G | 83.6 | CSwin-B | $384^2$ | 78M | 47.0G | 85.4 |
| InLine-CSwin-S | $224^2$ | 33M | 6.8G | 83.8 | InLine-CSwin-B | $384^2$ | 73M | 46.3G | 85.7 |

## 5.3 Empirical Analysis of Injectivity and Local Modeling

**Injective property**. As shown in Tab. 3, we adopt four different kernel function $\phi(\cdot)$ to validate the effect of injectivity. As discussed in Sec. 4.1, with kernel function $\phi(\cdot) = \mathrm{ReLU}(\cdot)$, linear attention fails to distinguish the same semantics with different intensities. Addressing this issue with InLine attention leads to a 2.5 increase in accuracy. When using $\phi(\cdot) = \mathrm{ReLU}(A \cdot + b)$, linear attention faces more severe semantic confusion, and introducing injective property results in a significant accuracy boost of 9.8, from 70.2 to 80.0. These obvious improvements fully prove the significance of injectivity and validate the effectiveness of our injective linear attention. We also employ two kernel functions that do not ensure non-negativity. Consistent with the findings in [27], linear attention fails to converge without non-negativity assurance. We attribute this to extreme semantic confusion. For example, with $\phi(\cdot) = \mathrm{Identity}(\cdot)$, linear attention is unable to distinguish completely opposite semantics, assigning identical attention scores to $q$ and $-q$.

**Local modeling capability**. Tab. 4 highlights the importance of local modeling capability. In the *left table*, we apply pure InLine attention to Swin-T and gradually increase the window size from $7^2$ to $56^2$. Due to the linear complexity of InLine attention, we can adopt different window sizes while preserving identical computational cost. Larger window sizes lead to larger receptive fields, typically associated with improved performance. However, the results show that the model performance does not improve with increasing window sizes. We believe this can be attributed to the insufficient local modeling capability: a small window size restricts the receptive field but introduces strong local bias, enhancing local modeling, while a large window size enlarges the receptive field but further diminishes local modeling ability. To validate this, we apply InLine attention with local residual and present the results in the *right table*. Significant improvements can be observed upon the introduction of the local residual term. Additionally, the increase in window size leads to steady performance improvement after introducing local residual, which strongly supports our analysis.

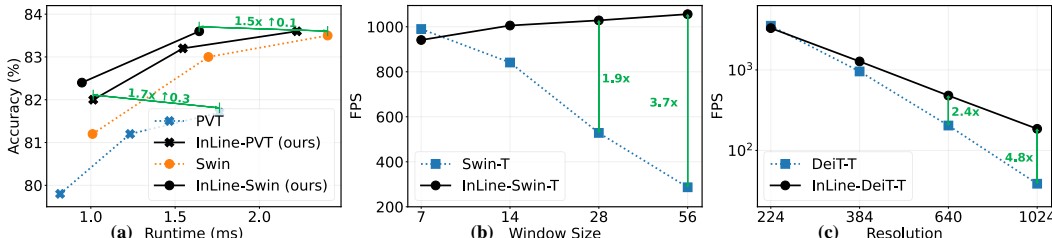

Figure 5: **Speed measurements**. Runtime and FPS is tested on a RTX3090 GPU. (a) Accuracy-Runtime curve on ImageNet. (b) Increasing window size. (c) High-resolution scenarios.

Table 7: Results on COCO dataset. The FLOPs are computed with an input resolution of $1280 \times 800$.

| **(a) Mask R-CNN Object Detection on COCO** | | | | | | | | |
|---|---|---|---|---|---|---|---|---|
| Method | FLOPs | Sch. | $AP^b$ | $AP^b_{50}$ | $AP^b_{75}$ | $AP^m$ | $AP^m_{50}$ | $AP^m_{75}$ |
| PVT-T | 240G | 1x | 36.7 | 59.2 | 39.3 | 35.1 | 56.7 | 37.3 |
| InLine-PVT-T | 211G | 1x | 40.2 | 62.7 | 43.8 | 37.7 | 59.7 | 40.4 |
| PVT-S | 305G | 1x | 40.4 | 62.9 | 43.8 | 37.8 | 60.1 | 40.3 |
| InLine-PVT-S | 250G | 1x | 43.4 | 66.4 | 47.1 | 40.1 | 63.1 | 43.3 |
| PVT-M | 392G | 1x | 42.0 | 64.4 | 45.6 | 39.0 | 61.6 | 42.1 |
| InLine-PVT-M | 310G | 1x | 44.0 | 66.4 | 48.0 | 40.3 | 63.4 | 43.5 |
| PVT-L | 494G | 1x | 42.9 | 65.0 | 46.6 | 39.5 | 61.9 | 42.5 |
| InLine-PVT-L | 377G | 1x | 45.4 | 67.6 | 49.7 | 41.4 | 64.7 | 44.6 |
| **(b) Cascade Mask R-CNN Object Detection on COCO** | | | | | | | | |
| Method | FLOPs | Sch. | $AP^b$ | $AP^b_{50}$ | $AP^b_{75}$ | $AP^m$ | $AP^m_{50}$ | $AP^m_{75}$ |
| Swin-S | 837G | 3x | 51.9 | 70.7 | 56.3 | 45.0 | 68.2 | 48.8 |
| InLine-Swin-S | 835G | 3x | 52.4 | 71.0 | 56.9 | 45.4 | 68.8 | 49.6 |
| Swin-B | 981G | 3x | 51.9 | 70.5 | 56.4 | 45.0 | 68.1 | 48.9 |
| InLine-Swin-B | 978G | 3x | 52.6 | 71.0 | 57.0 | 45.4 | 68.5 | 49.3 |

## 5.4 Main Results and Broad Comparisons

As shown in Tab. 4, InLine-Swin-T with local residual achieves better results than the Swin-T baseline, increasing from 81.3 to 82.4. Therefore, we wonder whether InLine attention module can perform better than the widely adopted Softmax attention in various scenarios. To validate this, we further apply it to several representative Transformers and conduct comprehensive comparisons on image classification, object detection, and semantic segmentation.

**ImageNet classification.** Firstly, We apply our InLine attention module to DeiT [30], PVT [33], and Swin Transformer [19], presenting the results in Tab. 5. It can be seen that substituting Softmax attention with our method results in notable improvements. For example, InLine-PVT-S outperforms PVT-L with 30% of the parameters and 40% of the FLOPs. Subsequently, We apply our module to the advanced Transformer design, CSwin Transformer [6], and offer a broad comparison with various state-of-the-art models on ImageNet-1K. As depicted in Tab. 6, our InLine-CSwin model not only yields better results than CSwin, but also surpasses various SOTA CNN and Transformer designs. These results demonstrate that our InLine attention module tends to be a superior alternative to the widely used Softmax attention.

**Inference throughput analysis.** We offer real speed measurements in Fig. 5. As shown in Fig. 5(a), InLine models achieve an obviously better trade-off between accuracy and latency. In Fig. 5(b), we increase the window size from $7^2$ to $56^2$. Due to the quadratic complexity of Softmax attention, Swin-T's speed drops sharply as window goes larger. On the contrary, InLine-Swin-T with linear complexity even exhibits higher speed with larger windows. This may be due to the reduction in the latency caused by the window partition. With a global receptive field, InLine benefits from both high performance (see Tab. 4) and fast speed. Furthermore, Fig. 5(c) shows the significant computational advantage of InLine in high-resolution scenarios.

**COCO object detection.** Tab. 7 shows that In-Line attention consistently improves the results in object detection tasks. For instance, InLine-PVT-S outperforms PVT-T with 6.7 box AP under similar FLOPs, and InLine-PVT-L surpasses PVT-M by 3.4 box AP with fewer FLOPs, showing the advantage of InLine attention's linear complexity in high-resolution scenarios.

**ADE20K semantic segmentation.** We employ our model on two representative segmentation models, SemanticFPN [16] and UperNet [37]. As depicted in Tab. 8, benefited from injectivity and effective local modeling ability, Our InLine achieves better results under all settings with obviously lower computational cost.

**Comparison with SOTA linear attention designs.** As shown in Tab. 9, our simple InLine attention design outperforms various linear attention methods without bells and whistles. Additionally, our Inline attention can possibly integrate with previous designs to achieve better results, which we leave for future work. For instance, the advanced focused function in FLatten [9] can also be employed in InLine attention.

Table 8: Results of semantic segmentation. The FLOPs are computed over encoders and decoders with an input image at the resolution of 512×2048. S-FPN is short for SemanticFPN [16] model.

| Semantic Segmentation on ADE20K | | | | | |
|---|---|---|---|---|---|
| Backbone | Method | FLOPs | #Params | mIoU | mAcc |
| PVT-T | S-FPN | 158G | 17M | 36.57 | 46.72 |
| InLine-PVT-T | S-FPN | 127G | 16M | **39.16** | **50.63** |
| PVT-S | S-FPN | 225G | 28M | 41.95 | 53.02 |
| InLine-PVT-S | S-FPN | 168G | 25M | **42.93** | **54.58** |
| PVT-L | S-FPN | 420G | 65M | 43.49 | 54.62 |
| InLine-PVT-L | S-FPN | 298G | 55M | **44.71** | **57.17** |
| Swin-T | UperNet | 945G | 60M | 44.51 | 55.61 |
| InLine-Swin-T | UperNet | 941G | 61M | **45.57** | **57.60** |

Table 9: Comparison of different linear attention designs using DeiT-T.

| Method | #Params | FLOPs | Acc. |
|---|---|---|---|
| Hydra Attn [1] | 5.7M | 1.1G | 68.3 |
| Efficient Attn [28] | 5.7M | 1.1G | 70.2 |
| Linear Angular Attn [40] | 5.7M | 1.1G | 70.8 |
| FLatten [9] | 6.1M | 1.1G | 74.1 |
| **InLine (Ours)** | 6.5M | 1.1G | **74.5** |

## 5.5 Ablation Study

The effectiveness of our two key designs has been verified and detailed analyzed in Sec. 5.3. In Tab. 10, we offer additional results to validate the impact of different kernel functions. It is shown that our InLine attention can effectively work with different kernel functions, further validating the effectiveness of our method. The ReLU and Exponential functions achieve slightly better results. In this paper, we use $\mathrm{Identity}(\cdot)$ as default for simplicity.

Table 10: Ablation on the impact of different kernel functions based on InLine-Swin-T.

| Kernel Function $\phi(\cdot)$ | Identity$(\cdot)$ | ReLU() | LeakyReLU$(\cdot)$ | Exponential$(\cdot)$ |
|---|---|---|---|---|
| Acc. | 82.4 | 82.5 | 82.3 | 82.5 |

## 6 Conclusion

In this paper, we shed some light on the core factors leading to the performance gap between linear and Softmax attention. We identify and validate two fundamental disparities between these two attention paradigms: injective property and local modeling capability. Injectivity implies that the attention function assigns distinct attention scores to queries with varying semantics, reflecting the ability to distinguish different semantics. Using different kernel functions, linear attention's non-injectivity results in various semantic confusions. Furthermore, despite being recognized for their robust long-range modeling capability, attention mechanisms heavily depend on effective local modeling for impressive results. Thorough empirical validation unequivocally supports our analyses. Our findings also demonstrate that with the above two properties, linear attention can outperform Softmax attention with lower computation complexity.

## Acknowledgement

This work is supported in part by the National Key R&D Program of China (2022ZD0114903).

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

# Appendix

## A  Mathematical Proof

This section offers mathematical proofs for the three propositions outlined in the main paper.

### A.1  Proof of Proposition 1

**Proposition 1** (Softmax attention is injective) *Given* $K \in \mathbb{R}^{N \times d}$ *with* $\text{rank}(K) = d$ *and* $\text{rank}([K, \mathbf{1}_{N \times 1}]) = d + 1$. $\forall\, p, q \in \mathbb{R}^d, p \neq q$, *we have* $\text{S}_\text{K}(p) \neq \text{S}_\text{K}(q)$.

*Proof.* To arrive at a contradiction, assume $\exists\, p, q \in \mathbb{R}^d, p \neq q$, s.t. $\text{S}_\text{K}(p) = \text{S}_\text{K}(q)$. Then we have:

$$\left[\exp(p^\top K_1), \cdots, \exp(p^\top K_N)\right]^\top \cdot \frac{\sum_{j=1}^{N} \exp(q^\top K_j)}{\sum_{j=1}^{N} \exp(p^\top K_j)} = \left[\exp(q^\top K_1), \cdots, \exp(q^\top K_N)\right]^\top \quad (7)$$

$$\Rightarrow \left[p^\top K_1, \cdots, p^\top K_N\right] + c = \left[q^\top K_1, \cdots, q^\top K_N\right], c = \ln\left[\frac{\sum_{j=1}^{N} \exp(q^\top K_j)}{\sum_{j=1}^{N} \exp(p^\top K_j)}\right] \quad (8)$$

Consider the following two cases:

1. $c = 0$. Then eq. (8) $\Rightarrow p^\top K_j = q^\top K_j \Rightarrow K(p - q) = 0$. As $\text{rank}(K) = d$, we have $p = q$, which contradicts the assumption that $p \neq q$.

2. $c \neq 0$. Then eq. (8) $\Rightarrow p^\top K_j + c = q^\top K_j \Rightarrow K_j^\top p + c = K_j^\top q \Rightarrow K_j^\top(p - q)/c = 1$. Therefore, we have $K_j^\top(p - q)/c = 1 \Rightarrow K(p - q)/c = \mathbf{1}_{N \times 1}$, which contradicts the fact that $\text{rank}([K, \mathbf{1}_{N \times 1}]) = d + 1$ and equation $Kx = \mathbf{1}_{N \times 1}$ does not have a solution.

Both cases arrive at a contradiction, which proves the original proposition. $\qquad\square$

*Notably, the two assumptions* $\text{rank}(K) = d$ *and* $\text{rank}([K, \mathbf{1}_{N \times 1}]) = d + 1$ *are easy to satisfy in real models.* In practice, the number of tokens $N$ is usually much larger than head dimension $d$. For instance, in the first stage of Swin Transformer [19], $N = 56^2 = 3136$ and $d = 32$. Therefore, we have $K \in \mathbb{R}^{N \times d}, N > d$. If $\text{rank}(K) < d$, it indicates that the $N$ key tokens lie in a low-dimension subspace of $\mathbb{R}^d$. If $\text{rank}([K, \mathbf{1}_{N \times 1}]) < d + 1$, it means that the $N$ key tokens are in a hyperplane of $\mathbb{R}^d$. Given that $N$ is much greater than $d$, it is highly likely that the $N$ key tokens have various directions in $\mathbb{R}^d$, instead of locating in a low-dimension subspace or hyperplane. In this case, it holds that $\text{rank}(K) = d$ and $\text{rank}([K, \mathbf{1}_{N \times 1}]) = d + 1$.

### A.2  Proof of Proposition 2

**Lemma 1** (Existence of collinear features in linear attention) *Let* $\phi : \mathbb{R}^d \to \mathbb{R}^d$ *be a continuous injective function.* $\exists p, q \in \mathbb{R}^d, p \neq q, \exists \alpha \in \mathbb{R}, \alpha \neq 0$, s.t. $\phi(q) = \alpha\phi(p)$.

*Proof.* Let $f : U \to S, f(x) = \frac{\phi(x)}{\|\phi(x)\|}, U = \{x \mid \phi(x) \neq 0, x \in \mathbb{R}^d\}, S = \{x \mid \|x\| = 1, x \in \mathbb{R}^d\}$. $\phi$ is a continuous function, so $f$ is continuous on $U$. $\phi$ is an injective function, so it has no more than one zero point. Therefore, $U = \mathbb{R}^d$ or $\{x \mid x \neq x_0, x \in \mathbb{R}^d\}$ is an open subset of $\mathbb{R}^d$, where $x_0$ is the zero point of $\phi$. $f(U) \in S$ is not an open set, as every point of $f(U)$ is not an interior point.

Assume $f$ is injective. Then $f : U \to S$ is a continuous injective map. According to *Brouwer Invariance of Domain Theorem*, $U$ is an open set $\Rightarrow f(U)$ is also open, which contradicts the fact that $f(U)$ is not an open set.

Therefore, $f$ is not injective. $\Rightarrow \exists p, q \in \mathbb{R}^d, p \neq q$, s.t. $f(p) = f(q)$.

$$\Rightarrow \phi(q) = \frac{\|\phi(q)\|}{\|\phi(p)\|}\phi(p) \triangleq \alpha\phi(p), \alpha = \frac{\|\phi(q)\|}{\|\phi(p)\|} \neq 0. \quad (9)$$

$\qquad\square$

**Proposition 2** (Linear attention is not injective) *Let $\phi : \mathbb{R}^d \to \mathbb{R}^d$ be a continuous function.* $\exists\, p, q \in \mathbb{R}^d, p \neq q$, s.t. $\mathrm{L_K}(p) = \mathrm{L_K}(q)$.

*Proof.* Consider the following two cases:

1. $\phi$ is not injective. $\exists p, q \in \mathbb{R}^d, p \neq q$, s.t. $\phi(p) = \phi(q) \Rightarrow \mathrm{L_K}(p) = \mathrm{L_K}(q)$.

2. $\phi$ is injective. Lemma $1 \Rightarrow \exists p, q \in \mathbb{R}^d, p \neq q, \exists \alpha \in \mathbb{R}, \alpha \neq 0$, s.t. $\phi(q) = \alpha\phi(p)$.

$$\Rightarrow \mathrm{L_K}(q) = \left[ \frac{\alpha\phi(p)^\top \phi(K_1)}{\sum_{j=1}^N \alpha\phi(p)^\top \phi(K_j)}, \cdots, \frac{\alpha\phi(p)^\top \phi(K_N)}{\sum_{j=1}^N \alpha\phi(p)^\top \phi(K_j)} \right]^\top$$

$$= \left[ \frac{\phi(p)^\top \phi(K_1)}{\sum_{j=1}^N \phi(p)^\top \phi(K_j)}, \cdots, \frac{\phi(p)^\top \phi(K_N)}{\sum_{j=1}^N \phi(p)^\top \phi(K_j)} \right]^\top = \mathrm{L_K}(p). \tag{10}$$

$\square$

### A.3 Proof of Proposition 3

**Proposition 3** (InLine attention is injective) *Let $\phi : \mathbb{R}^d \to \mathbb{R}^d$ be an injective map. Given $K \in \mathbb{R}^{N \times d}$ with* $\mathrm{rank}(\phi(K)) = d$ *and* $\mathrm{rank}([\phi(K), \mathbf{1}_{N \times 1}]) = d + 1$. $\forall p, q \in \mathbb{R}^d, p \neq q, \Rightarrow \mathrm{InL_K}(p) \neq \mathrm{InL_K}(q)$.

*Proof.* Assume $\exists\, p, q \in \mathbb{R}^d, p \neq q$, s.t. $\mathrm{InL_K}(p) = \mathrm{InL_K}(q)$. $p \neq q \Rightarrow \phi(p) \neq \phi(q)$.

$$\Rightarrow \left[ \phi(p)^\top \phi(K_1), \cdots, \phi(p)^\top \phi(K_N) \right] + c = \left[ \phi(q)^\top \phi(K_1), \cdots, \phi(q)^\top \phi(K_N) \right],$$

$$c = \frac{1}{N} \sum_{j=1}^N \phi(q)^\top \phi(K_j) - \frac{1}{N} \sum_{j=1}^N \phi(p)^\top \phi(K_j). \tag{11}$$

The subsequent proof mirrors Proposition 1. $\square$

*Similar to the analysis in Sec. A.1,* $\mathrm{rank}(\phi(K)) = d$ *and* $\mathrm{rank}([\phi(K), \mathbf{1}_{N \times 1}]) = d + 1$ *are easy to satisfy.*

## B   Complete Experimental Results

We provide complete experimental results on ImageNet-1K classification [5], COCO object detection [18], ADE20K semantic segmentation [44] in Tab. 11, Tab. 12 and Tab. 13. The results demonstrate that InLine attention module consistently outperforms Softmax counterparts across all settings, strongly supporting our analyses and fully validating its effectiveness.

## C   Model Architectures

We summarize the detailed architectures of four Transformer models used in the main paper, including InLine-DeiT, InLine-PVT, InLine-Swin and InLine-CSwin in Tab.14-18.

## D   Limitations

In this paper, we shed some light on the core factors leading to the performance gap between linear and Softmax attention. We identify and validate two fundamental and essential disparities between these two attention paradigms: injective property and local modeling capability. Firstly, we prove that linear attention is not injective, which is prone to assign identical attention weights to different query vectors, thus adding to severe semantic confusion problem. Secondly, we confirm that effective local modeling is important for the success of Softmax attention, in which linear attention falls short. The aforementioned two essential differences significantly contribute to the disparities between these two attention paradigms, which is unequivocally proved by our thorough empirical validation in the paper. However, there may be other differences between Softmax and linear attention, and this paper is not exhaustive.

Table 11: Comparison with baseline models on ImageNet-1K.

| Method | Reso | #Params | FLOPs | Top-1 |
|--------|------|---------|-------|-------|
| DeiT-T [30] | $224^2$ | 5.7M | 1.2G | 72.2 |
| **InLine-DeiT-T** | $224^2$ | 6.5M | 1.1G | **74.5** (+2.3) |
| DeiT-S | $224^2$ | 22.1M | 4.6G | 79.8 |
| **InLine-DeiT-S** | $224^2$ | 16.7M | 5.0G | **80.2** (+0.4) |
| DeiT-B | $224^2$ | 86.6M | 17.6G | 81.8 |
| **InLine-DeiT-B** | $448^2$ | 23.8M | 17.2G | **82.3** (+0.5) |
| PVT-T [33] | $224^2$ | 13.2M | 1.9G | 75.1 |
| **InLine-PVT-T** | $224^2$ | 12.0M | 2.0G | **78.2** (+3.1) |
| PVT-S | $224^2$ | 24.5M | 3.8G | 79.8 |
| **InLine-PVT-S** | $224^2$ | 21.6M | 3.9G | **82.0** (+2.2) |
| PVT-M | $224^2$ | 44.2M | 6.7G | 81.2 |
| **InLine-PVT-M** | $224^2$ | 37.6M | 6.9G | **83.2** (+2.0) |
| PVT-L | $224^2$ | 61.4M | 9.8G | 81.7 |
| **InLine-PVT-L** | $224^2$ | 50.2M | 10.2G | **83.6** (+1.9) |
| Swin-T [19] | $224^2$ | 29M | 4.5G | 81.3 |
| **InLine-Swin-T** | $224^2$ | 30M | 4.5G | **82.4** (+1.1) |
| Swin-S | $224^2$ | 50M | 8.7G | 83.0 |
| **InLine-Swin-S** | $224^2$ | 50M | 8.7G | **83.6** (+0.6) |
| Swin-B | $224^2$ | 88M | 15.4G | 83.5 |
| **InLine-Swin-B** | $224^2$ | 88M | 15.4G | **84.1** (+0.6) |
| Swin-B | $384^2$ | 88M | 47.0G | 84.5 |
| **InLine-Swin-B** | $384^2$ | 88M | 45.2G | **85.0** (+0.5) |

Table 12: Results on COCO dataset. The FLOPs are computed with an input resolution of $1280\times800$.

| (a) Mask R-CNN Object Detection on COCO | | | | | | | | |
|---------|-------|------|--------|-------------|-------------|--------|-------------|-------------|
| Method | FLOPs | Sch. | $AP^b$ | $AP^b_{50}$ | $AP^b_{75}$ | $AP^m$ | $AP^m_{50}$ | $AP^m_{75}$ |
| PVT-T | 240G | 1x | 36.7 | 59.2 | 39.3 | 35.1 | 56.7 | 37.3 |
| InLine-PVT-T | 211G | 1x | 40.2 | 62.7 | 43.8 | 37.7 | 59.7 | 40.4 |
| PVT-S | 305G | 1x | 40.4 | 62.9 | 43.8 | 37.8 | 60.1 | 40.3 |
| InLine-PVT-S | 250G | 1x | 43.4 | 66.4 | 47.1 | 40.1 | 63.1 | 43.3 |
| PVT-M | 392G | 1x | 42.0 | 64.4 | 45.6 | 39.0 | 61.6 | 42.1 |
| InLine-PVT-M | 310G | 1x | 44.0 | 66.4 | 48.0 | 40.3 | 63.4 | 43.5 |
| PVT-L | 494G | 1x | 42.9 | 65.0 | 46.6 | 39.5 | 61.9 | 42.5 |
| InLine-PVT-L | 377G | 1x | 45.4 | 67.6 | 49.7 | 41.4 | 64.7 | 44.6 |
| (b) Cascade Mask R-CNN Object Detection on COCO | | | | | | | | |
| Method | FLOPs | Sch. | $AP^b$ | $AP^b_{50}$ | $AP^b_{75}$ | $AP^m$ | $AP^m_{50}$ | $AP^m_{75}$ |
| Swin-S | 837G | 3x | 51.9 | 70.7 | 56.3 | 45.0 | 68.2 | 48.8 |
| InLine-Swin-S | 835G | 3x | 52.4 | 71.0 | 56.9 | 45.4 | 68.8 | 49.6 |
| Swin-B | 981G | 3x | 51.9 | 70.5 | 56.4 | 45.0 | 68.1 | 48.9 |
| InLine-Swin-B | 978G | 3x | 52.6 | 71.0 | 57.0 | 45.4 | 68.5 | 49.3 |

Table 13: Results of semantic segmentation. The FLOPs are computed over encoders and decoders with an input image at the resolution of 512×2048. S-FPN is short for SemanticFPN [16] model.

| Semantic Segmentation on ADE20K | | | | | |
|---|---|---|---|---|---|
| Backbone | Method | FLOPs | #Params | mIoU | mAcc |
| PVT-T | S-FPN | 158G | 17M | 36.57 | 46.72 |
| InLine-PVT-T | S-FPN | 127G | 16M | **39.16** | **50.63** |
| PVT-S | S-FPN | 225G | 28M | 41.95 | 53.02 |
| InLine-PVT-S | S-FPN | 168G | 25M | **42.93** | **54.58** |
| PVT-M | S-FPN | 315G | 48M | 42.91 | 53.80 |
| InLine-PVT-M | S-FPN | 229G | 41M | **44.59** | **57.20** |
| PVT-L | S-FPN | 420G | 65M | 43.49 | 54.62 |
| InLine-PVT-L | S-FPN | 298G | 55M | **44.71** | **57.17** |
| Swin-T | UperNet | 945G | 60M | 44.51 | 55.61 |
| InLine-Swin-T | UperNet | 941G | 61M | **45.57** | **57.60** |
| Swin-S | UperNet | 1038G | 81M | 47.64 | 58.78 |
| InLine-Swin-S | UperNet | 1035G | 81M | **48.59** | **60.73** |
| Swin-B | UperNet | 1188G | 121M | 48.13 | 59.13 |
| InLine-Swin-B | UperNet | 1183G | 122M | **49.10** | **60.57** |

Table 14: Architectures of InLine-DeiT models.

| InLine-DeiT-T | | InLine-DeiT-S | | InLine-DeiT-B | |
|---|---|---|---|---|---|
| InLine Block | DeiT Block | InLine Block | DeiT Block | InLine Block | DeiT Block |
| $\begin{bmatrix} \text{res } 14\times14 \\ \text{dim } 192 \\ \text{head } 6 \end{bmatrix} \times 12$ | None | $\begin{bmatrix} \text{res } 18\times18 \\ \text{dim } 320 \\ \text{head } 10 \end{bmatrix} \times 12$ | None | $\begin{bmatrix} \text{res } 28\times28 \\ \text{dim } 384 \\ \text{head } 12 \end{bmatrix} \times 12$ | None |

Table 15: Architectures of InLine-PVT models (Part1).

| stage | output | InLine-PVT-T | | InLine-PVT-S | |
|---|---|---|---|---|---|
| | | InLine Block | PVT Block | InLine Block | PVT Block |
| res1 | 56 × 56 | Conv4×4, stride=4, 64, LN | | | |
| | | $\begin{bmatrix} \text{win } 56\times56 \\ \text{dim } 64 \\ \text{head } 1 \end{bmatrix} \times 2$ | None | $\begin{bmatrix} \text{win } 56\times56 \\ \text{dim } 64 \\ \text{head } 1 \end{bmatrix} \times 3$ | None |
| res2 | 28 × 28 | Conv2×2, stride=2, 128, LN | | | |
| | | $\begin{bmatrix} \text{win } 28\times28 \\ \text{dim } 128 \\ \text{head } 2 \end{bmatrix} \times 2$ | None | $\begin{bmatrix} \text{win } 28\times28 \\ \text{dim } 128 \\ \text{head } 2 \end{bmatrix} \times 3$ | None |
| res3 | 14 × 14 | Conv2×2, stride=2, 320, LN | | | |
| | | $\begin{bmatrix} \text{win } 14\times14 \\ \text{dim } 320 \\ \text{head } 5 \end{bmatrix} \times 2$ | None | $\begin{bmatrix} \text{win } 14\times14 \\ \text{dim } 320 \\ \text{head } 5 \end{bmatrix} \times 6$ | None |
| res4 | 7 × 7 | Conv2×2, stride=2, 512, LN | | | |
| | | $\begin{bmatrix} \text{win } 7\times7 \\ \text{dim } 512 \\ \text{head } 8 \end{bmatrix} \times 2$ | None | $\begin{bmatrix} \text{win } 7\times7 \\ \text{dim } 512 \\ \text{head } 8 \end{bmatrix} \times 3$ | None |

Table 16: Architectures of InLine-PVT models (Part2).

| stage | output | InLine-PVT-M | | InLine-PVT-L | |
|---|---|---|---|---|---|
| | | InLine Block | PVT Block | InLine Block | PVT Block |
| res1 | $56 \times 56$ | Conv4×4, stride=4, 64, LN | | | |
| | | $\begin{bmatrix} \text{win } 56\times56 \\ \text{dim } 64 \\ \text{head } 1 \end{bmatrix} \times 3$ | None | $\begin{bmatrix} \text{win } 56\times56 \\ \text{dim } 64 \\ \text{head } 1 \end{bmatrix} \times 3$ | None |
| res2 | $28 \times 28$ | Conv2×2, stride=2, 128, LN | | | |
| | | $\begin{bmatrix} \text{win } 28\times28 \\ \text{dim } 128 \\ \text{head } 2 \end{bmatrix} \times 3$ | None | $\begin{bmatrix} \text{win } 28\times28 \\ \text{dim } 128 \\ \text{head } 2 \end{bmatrix} \times 8$ | None |
| res3 | $14 \times 14$ | Conv2×2, stride=2, 320, LN | | | |
| | | $\begin{bmatrix} \text{win } 14\times14 \\ \text{dim } 320 \\ \text{head } 5 \end{bmatrix} \times 18$ | None | $\begin{bmatrix} \text{win } 14\times14 \\ \text{dim } 320 \\ \text{head } 5 \end{bmatrix} \times 27$ | None |
| res4 | $7 \times 7$ | Conv2×2, stride=2, 512, LN | | | |
| | | $\begin{bmatrix} \text{win } 7\times7 \\ \text{dim } 512 \\ \text{head } 8 \end{bmatrix} \times 3$ | None | $\begin{bmatrix} \text{win } 7\times7 \\ \text{dim } 512 \\ \text{head } 8 \end{bmatrix} \times 3$ | None |

Table 17: Architectures of InLine-Swin models.

| stage | output | InLine-Swin-T | | InLine-Swin-S | | InLine-Swin-B | |
|---|---|---|---|---|---|---|---|
| | | InLine Block | Swin Block | InLine Block | Swin Block | InLine Block | Swin Block |
| res1 | $56 \times 56$ | concat $4 \times 4$, 96, LN | | concat $4 \times 4$, 96, LN | | concat $4 \times 4$, 128, LN | |
| | | $\begin{bmatrix} \text{win } 56\times56 \\ \text{dim } 96 \\ \text{head } 3 \end{bmatrix} \times 2$ | None | $\begin{bmatrix} \text{win } 56\times56 \\ \text{dim } 96 \\ \text{head } 3 \end{bmatrix} \times 2$ | None | $\begin{bmatrix} \text{win } 56\times56 \\ \text{dim } 128 \\ \text{head } 3 \end{bmatrix} \times 2$ | None |
| res2 | $28 \times 28$ | concat $2 \times 2$, 192, LN | | concat $2 \times 2$, 192, LN | | concat $2 \times 2$, 256, LN | |
| | | $\begin{bmatrix} \text{win } 28\times28 \\ \text{dim } 192 \\ \text{head } 6 \end{bmatrix} \times 2$ | None | $\begin{bmatrix} \text{win } 28\times28 \\ \text{dim } 192 \\ \text{head } 6 \end{bmatrix} \times 2$ | None | $\begin{bmatrix} \text{win } 28\times28 \\ \text{dim } 256 \\ \text{head } 6 \end{bmatrix} \times 2$ | None |
| res3 | $14 \times 14$ | concat $2 \times 2$, 384, LN | | concat $2 \times 2$, 384, LN | | concat $2 \times 2$, 512, LN | |
| | | None | $\begin{bmatrix} \text{win } 7\times7 \\ \text{dim } 384 \\ \text{head } 12 \end{bmatrix} \times 6$ | None | $\begin{bmatrix} \text{win } 7\times7 \\ \text{dim } 384 \\ \text{head } 12 \end{bmatrix} \times 18$ | $\begin{bmatrix} \text{win } 14\times14 \\ \text{dim } 512 \\ \text{head } 12 \end{bmatrix} \times 2$ | $\begin{bmatrix} \text{win } 7\times7 \\ \text{dim } 512 \\ \text{head } 12 \end{bmatrix} \times 16$ |
| res4 | $7 \times 7$ | concat $2 \times 2$, 768, LN | | concat $2 \times 2$, 768, LN | | concat $2 \times 2$, 1024, LN | |
| | | None | $\begin{bmatrix} \text{win } 7\times7 \\ \text{dim } 768 \\ \text{head } 24 \end{bmatrix} \times 2$ | None | $\begin{bmatrix} \text{win } 7\times7 \\ \text{dim } 768 \\ \text{head } 24 \end{bmatrix} \times 2$ | None | $\begin{bmatrix} \text{win } 7\times7 \\ \text{dim } 1024 \\ \text{head } 24 \end{bmatrix} \times 2$ |

Table 18: Architectures of InLine-CSwin models.

| stage | output | InLine-CSwin-T | | InLine-CSwin-S | | InLine-CSwin-B | |
|---|---|---|---|---|---|---|---|
| | | InLine Block | CSwin Block | InLine Block | CSwin Block | InLine Block | CSwin Block |
| res1 | 56 × 56 | Conv7×7, stride=4, 64, LN | | Conv7×7, stride=4, 64, LN | | Conv7×7, stride=4, 96, LN | |
| | | $\begin{bmatrix} \text{win } 56\times56 \\ \text{dim } 64 \\ \text{head } 2 \end{bmatrix} \times 2$ | None | $\begin{bmatrix} \text{win } 56\times56 \\ \text{dim } 64 \\ \text{head } 2 \end{bmatrix} \times 3$ | None | $\begin{bmatrix} \text{win } 56\times56 \\ \text{dim } 96 \\ \text{head } 4 \end{bmatrix} \times 3$ | None |
| res2 | 28 × 28 | Conv3×3, stride=2, 128, LN | | Conv3×3, stride=2, 128, LN | | Conv3×3, stride=2, 192, LN | |
| | | $\begin{bmatrix} \text{win } 28\times28 \\ \text{dim } 128 \\ \text{head } 4 \end{bmatrix} \times 4$ | None | $\begin{bmatrix} \text{win } 28\times28 \\ \text{dim } 128 \\ \text{head } 4 \end{bmatrix} \times 6$ | None | $\begin{bmatrix} \text{win } 28\times28 \\ \text{dim } 192 \\ \text{head } 8 \end{bmatrix} \times 6$ | None |
| res3 | 14 × 14 | Conv3×3, stride=2, 256, LN | | Conv3×3, stride=2, 256, LN | | Conv3×3, stride=2, 384, LN | |
| | | None | $\begin{bmatrix} \text{win } 7\times14 \\ \text{dim } 256 \\ \text{head } 8 \end{bmatrix} \times 18$ | None | $\begin{bmatrix} \text{win } 7\times14 \\ \text{dim } 256 \\ \text{head } 8 \end{bmatrix} \times 29$ | None | $\begin{bmatrix} \text{win } 7\times14 \\ \text{dim } 384 \\ \text{head } 16 \end{bmatrix} \times 29$ |
| res4 | 7 × 7 | Conv3×3, stride=2, 512, LN | | Conv3×3, stride=2, 512, LN | | Conv3×3, stride=2, 768, LN | |
| | | None | $\begin{bmatrix} \text{win } 7\times7 \\ \text{dim } 512 \\ \text{head } 16 \end{bmatrix} \times 1$ | None | $\begin{bmatrix} \text{win } 7\times7 \\ \text{dim } 512 \\ \text{head } 16 \end{bmatrix} \times 2$ | None | $\begin{bmatrix} \text{win } 7\times7 \\ \text{dim } 768 \\ \text{head } 32 \end{bmatrix} \times 2$ |

