# OpenReview forum: "Bridging the Divide: Reconsidering Softmax and Linear Attention"
_NeurIPS.cc/2024/Conference — NeurIPS 2024 poster_

### Official Review · Reviewer_yvdC · 2024-06-28

**Soundness:** 3
**Presentation:** 3
**Contribution:** 3
**Rating:** 6
**Confidence:** 4

**Summary:**

The paper addresses the computational inefficiency of Softmax attention in Vision Transformers, particularly when handling high-resolution inputs. The authors provide a theoretical analysis showing that the injectivity and local modeling capabilities of attention mechanisms significantly impact performance. They demonstrate that linear attention, which has linear complexity, is not injective and thus performs poorly compared to Softmax attention. To address this, the authors propose modifications to make linear attention injective, resulting in InLine Attention, which improves performance in vision tasks while maintaining computational efficiency. Experiments on high-resolution vision tasks show that InLine attention exhibits comparable performance to Softmax attention.

**Strengths:**

- The paper presents a solid theoretical analysis explaining the performance gap between linear and Softmax attention, focusing on injectivity and local modeling capabilities.
- The proposed modifications to linear attention are simple yet improve the performance and computational efficiency of Vision Transformers.

**Weaknesses:**

- The only novelty in the paper is the analysis of the injective property of Softmax and linear attention. The overall novelty and contribution of the paper are limited.
- The analysis of the local modeling capability of Softmax and linear attention was originally presented by [1]. The authors do not mention this work in their manuscript.
- The authors do not mention or provide any ablation study of the different embedding functions that can be used with InLine attention. Some works suggest that the exponential function is more beneficial than ReLU [2]. If the authors can show that InLine attention has similar performance for both ReLU and exponential functions, it could justify the strength of the method.
- There is a lack of experiments on language models. Generally, language models are much harder to train with linear attention than Vision Transformers. Vision tasks mostly require learning local interactions where language exhibits more long-range dependencies between tokens. If the authors can show their method works on language models, it could improve the soundness of this work.

**Questions:**

Normalizing linear attention by its mean (centering) as suggested in eq. 4 should result in negative attention scores. To be strictly positive, it requires all numbers to be greater than $-1/N$. Can the authors provide some analysis or ablation study showing the effect of negative values on the attention on performance?

[1] The Devil in Linear Transformer

[2] Linear Log-Normal Attention with Unbiased Concentration

---

> ### Author Rebuttal · Authors · 2024-08-06
>
> We would first like to express our appreciation for your time and insightful comments. Please find our response to your concerns in the following:
>
> ---
>
> **1. Novelty and contribution.**
>
> Thanks for your valuable comment. The novelty and contribution of our work can be summarized as follows:
>
> - We identify an important property of attention mechanism, ***injectivity***, which has not been explored in literature. With both theoretical proof and experimental verification, we show that injectivity significantly contributes to the performance gap between Softmax and linear attention.
> - We thoroughly validate the crucial role of ***local modeling*** in attention mechanism through a series of experiments.
> - We propose a novel ***subtraction normalization*** method to achieve injectivity in linear attention, significantly improving performance with no extra FLOPs.
> - We present ***InLine attention module***, a simple, effective and efficient alternative to the commonly adopted Softmax attention. Extensive experiments fully validate the superiority of our design over Softmax attention.
>
> ---
>
> **2. The locality problem and related work.**
>
> Thanks for highlighting this important related work [1]. While our work shares some similar analyses with [1], it has fundamental distinctions:
>
> - ***We thoroughly validate the crucial role of local modeling in attention mechanism, while [1] mainly focuses on identifying differences.*** There could be many different behaviors between Softmax and linear attention, but a large number of them may not be the key disparities. Therefore, our study conducts a series of experiments in Fig. 3, 4 and Table. 2, 4 to fully validate that: (1) Softmax attention's effectiveness depends significantly on robust local modeling. (2) Varied local priors contribute substantially to the performance gap between Softmax and linear attention. In contrast, [1] observes that Softmax attention is more concentrated locally than linear attention (see its Fig. 2), but it does not verify whether this is a key factor leading to the performance gap between the two.
>
> - ***The design proposed in our paper is more effective than [1].*** To enhance locality, [1] proposes DiagAttention, which restricts each token's receptive field to its local block. This approach resembles the window attention in Table 4 (left), which shows little improvement (window=$14^2$, acc=80.4) compared to vanilla global linear attention (acc=80.2). In contrast, our design boosts accuracy significantly from 80.2 to 82.4, fully unleashing the power of global linear attention. Additionally, we offer a comparison between our InLine model and TransNormer proposed in [1] under Swin-T structure. Our InLine model achieves better results.
>
>   |   Method    | #Params | FLOPs | Acc. |
>   | :---------: | :-----: | :---: | :--: |
>   | TransNormer |   30M   | 4.5G  | 79.9 |
>   |   InLine    |   30M   | 4.5G  | 82.4 |
>
> - Thanks for pointing this out again. ***We will include discussion and give more credits to this work in the revised manuscript.***
>
> [1] The Devil in Linear Transformer.
>
> ---
>
> **3. Ablation on different kernel functions.**
>
> Thanks for the insightful comment. We conduct additional ablation studies on different kernel functions $\phi$ using InLine-Swin-T.
>
> | Kernel Function $\phi$ | #Params | FLOPs | Acc. |
> | :--------------------: | :-----: | :---: | :--: |
> |        Identity        |   30M   | 4.5G  | 82.4 |
> |          ReLu          |   30M   | 4.5G  | 82.5 |
> |       LeakyReLu        |   30M   | 4.5G  | 82.3 |
> |      Exponential       |   30M   | 4.5G  | 82.5 |
>
> It is shown that our InLine attention can effectively work with different kernel functions, further validating the effectiveness of our method. The ReLU and Exponential functions achieve similar results, slightly outperforming the Identity function. In our paper, we use Identity kernel function as default. Additionally, we will give more credits to [2] in the revised version.
>
> [2] Linear Log-Normal Attention with Unbiased Concentration
>
> ---
>
> **4. Experiments on language models.**
>
> Please refer to our general response.
>
> ---
>
> **5. Negative attention scores.**
>
> Thanks for the insightful question. We already provide a brief discussion on the non-negativity assurance in L260-L263 of our paper. Here, we offer detailed analyses and additional experiments to further clarify this interesting issue.
>
> - ***InLine attention does not ensure non-negative attention scores.*** As you pointed out, the subtraction normalization could possibly produce negative values, and we practically find that there do exist many negative attention scores in our InLine models.
>
> - ***These negative values do not impair the performance of InLine attention.*** We directly verify this with an ablation experiment, where we ensure all attention values to be non-negative by introducing additional normalization on $Q,K$. The results are provided below.
>
>   |           Method           | #Params | FLOPs | Acc. |
>   | :------------------------: | :-----: | :---: | :--: |
>   |       InLine-Swin-T        |   30M   | 4.5G  | 82.4 |
>   | InLine-Swin-T non-negative |   30M   | 4.5G  | 81.7 |
>
>   It can be seen that the additional non-negative assurance actually leads to notable performance drop. This suggests that negative values in InLine attention do not hinder performance and might even enhance the model.
>
> - ***Non-negativity is crucial for vanilla linear attention.*** As depicted in Table. 3 of our paper, unlike our InLine attention, vanilla linear attention fails to converge without non-negativity assurance. We attribute this to extreme non-injectivity and semantic confusion problem. For example, with Identity kernel function, vanilla linear attention is unable to distinguish completely opposite semantics, assigning identical attention scores to $q$ and $−q$.

---

> > ### Comment · Reviewer_yvdC · 2024-08-10
> >
> > Thank you for the rebuttal. The authors have addressed my request for additional ablation of kernel functions and provided further experimental results. Based on this additional information, I have decided to increase my score.

---

> > > ### Author Response · Authors · 2024-08-12
> > >
> > > Thank you for your time and valuable comments. If there are any additional questions or concerns, we are more than happy to provide further clarification to fully address them.

---

### Official Review · Reviewer_3cj2 · 2024-07-12

**Soundness:** 3
**Presentation:** 3
**Contribution:** 3
**Rating:** 5
**Confidence:** 4

**Summary:**

This paper invetigate the linear attention in the Vision task

**Strengths:**

1. The paper is well written, the motivation is clear.

2. The findings is this work is 1) inear attention is not injective, which is prone to assign identical attention weights to different query vector. 2) effective local modeling is essential for the success of Softmax attention, which linear attention deos not have. Thoses findings may be improtant for design efficient linear transformer.

3. The experimental results are good.

**Weaknesses:**

1. The source code is not avaliable, the reproductability is unclear at this time.

2. how this linear attetnion compare with Vision Mamba since Mamba is efficient.

3. The experimental results are mainly on CV domain, is this algorithm adaptive to NLP or time series domain?

**Questions:**

1. The source code is not avaliable, the reproductability is unclear at this time.

2. how this linear attetnion compare with Vision Mamba since Mamba is efficient.

3. The experimental results are mainly on CV domain, is this algorithm adaptive to NLP or time series domain?

**Limitations:**

The experimental results are mainly on CV domain, is this algorithm adaptive to NLP or time series domain?

---

> ### Author Rebuttal · Authors · 2024-08-06
>
> We would first like to express our appreciation for your time and insightful comments. Please find our response to your concerns in the following:
>
> ---
>
> **1. The source code.**
>
> - ***Firstly, we provide the Pytorch-style pseudo code of our InLine attention module below.*** It can be seen that the proposed method is very simple and easy to implement.
>
>   ```python
>   # b: batch size;  n: sequence length;  d: head dimension
>
>   # q: query,		shape: (b, num_heads, n, d)
>   # k: key,		shape: (b, num_heads, n, d)
>   # v: value,		shape: (b, num_heads, n, d)
>
>   # Eq. 5 of our paper
>   o = q @ (k.transpose(2, 3) @ v) - (q @ k.transpose(2, 3).sum(dim=2) - 1) * v.mean(dim=2)
>   # Eq. 6 of our paper
>   o = o + local_residual_aggre(v)
>
>   # o: output,	shape: (b, num_heads, n, d)
>   ```
>
> - ***Secondly, we will provide the full source code to promote reproducibility if this manuscript is accepted.***
>
> ---
>
> **2. Comparison with vision Mamba models.**
>
> Thanks for your valuable comment. Here, we provide additional comparison with vision Mamba models.
>
> - ***We offer comparison in terms of params, FLOPs and accuracy in the table below.***
>
>   |       Model        | #Params |   FLOPs   |   Acc.   |
>   | :----------------: | :-----: | :-------: | :------: |
>   |       Vim-S        |   26M   |   5.1G    |   80.3   |
>   |     LocalVim-S     |   28M   |   4.8G    |   81.2   |
>   |   PlainMamba-L2    |   25M   |   8.1G    |   81.6   |
>   | EfficientVMamba-B  |   33M   |   4.0G    |   81.8   |
>   |      VMamba-T      |   31M   |   4.9G    |   82.5   |
>   |   LocalVMamba-T    |   26M   |   5.7G    |   82.7   |
>   | **InLine-CSwin-T** | **21M** | **4.3G**  | **83.2** |
>   |                    |         |           |          |
>   |     Mamba2D-B      |   94M   |     -     |   83.0   |
>   |      VMamba-B      |   89M   |   15.4G   |   83.9   |
>   | **InLine-CSwin-B** | **73M** | **14.9G** | **84.5** |
>
>   As depicted in the above table, our InLine model surpasses various vision Mamba designs without bells and whistles.
>
> - ***Furthermore, as our InLine attention is extremely simple, it achieves much faster inference speed than vision Mamba models.*** Accuracy-runtime comparison is provided ***in the PDF in general response***. The results demonstrate that our InLine model is ***4.6x, 6.3x faster*** than LocalVMamba and PlainMamba models respectively, while achieving comparable performance. Compared to the highly optimized VMamba model, our model also achieves 1.4x speedup and 0.2 accuracy gain.
>
> ---
>
> **3. Is this algorithm adaptive to NLP or time series domain?**
>
> Our method can be applied to NLP and time series domain. Please refer to our general response for the detailed discussion.

---

> ### Author Response · Authors · 2024-08-12
>
> Dear Reviewer 3cj2, thank you for your insightful review and for engaging with our work. We would like to know if there are any additional questions or concerns. We are eager to engage in further discussion and provide clarification to fully address them.

---

### Official Review · Reviewer_My8Q · 2024-07-14

**Soundness:** 3
**Presentation:** 3
**Contribution:** 2
**Rating:** 5
**Confidence:** 3

**Summary:**

This paper aims to solve the computational challenges of Softmax attention in vision tasks due to its quadratic complexity with respect to sequence length. Linear attention as an alternative, reduces complexity to linear time by altering the similarity function from Softmax to kernel functions. However, the authors argue linear attention’s poor expressive power and non-injective nature can lead to semantic confusion. The authors propose two methods to enhance linear attention: enforcing injective properties and improving local modeling capabilities. Using the Swin Transformer architecture, they validate these methods, showing that linear attention can match or exceed Softmax attention’s performance while maintaining lower computational costs. The main contributions are highlighting the importance of injectivity and local modeling in attention mechanisms and demonstrating that linear attention, with these enhancements, can outperform traditional Softmax attention.

**Strengths:**

1.	This paper thoroughly analyzes the shortcomings of linear attention in vision tasks compared to Softmax attention, identifying non-injective properties and attention confusion as potential root causes. The authors validate these issues through quantitative and qualitative experiments, demonstrating that they contribute to performance drops. The claims seem well-founded, and the verification process appears robust.

2.	To address these issues, the authors propose a simple yet effective modification: using subtraction in the normalization of linear attention instead of division, creating a method they call injective linear attention (InLine).

3.	The proposed InLine method achieves competitive performance on ImageNet 1k classification and various downstream tasks.

**Weaknesses:**

1. Although the authors' hypothesis and claims seem reasonable, the performance of the proposed method is not remarkable. This paper also lacks the comparison to some related works. For example, another linear attention based method VVT [A] achieves the Top-1 Acc(%) of 84.1 on ImageNet1k with 61.8M Param and 10.8 GFLOPs. The proposed method, InLine-CSwin-B has a Top-1 Acc(%) of 84.5 on ImageNet1k with 73M Param and 14.9G FLOPs. Although InLine-CSwin-B is higher regarding accuracy by 0.4%, it uses 20% more Params and 40% more GFLOPs. This largely weakens the authors' claims.

2. The analysis of local modeling capability (L264-L275) indicates correlation rather than causation. The authors gradually increase the window size and find it does not lead to better performance. Based on this observation, they claim “the insufficient local modeling capability: a small window size restricts the receptive field but introduces strong local bias, enhancing local modeling, while a large window size enlarges the receptive field but further diminishes local modeling ability”. They find that adding a residual connection can solve the problem and thus claim that the local modeling capability of linear attention is problematic. However, as a common practice, a larger receptive field usually requires a different learning rate to ensure the network converges sufficiently. The effect of the residual connection here may not be as the authors claim, but rather just stabilizing the gradient.


[A] Sun, W., Qin, Z., Deng, H., Wang, J., Zhang, Y., Zhang, K., Barnes, N., Birchfield, S., Kong, L. and Zhong, Y., 2023. Vicinity vision transformer. IEEE Transactions on Pattern Analysis and Machine Intelligence, 45(10), pp.12635-12649.

**Questions:**

Overall the reviewer is quite concerned about the performance of the proposed method, especially given the fact it loses the direct comparison to a very related and comparable linear attention model.

**Limitations:**

Yes.

---

> ### Author Rebuttal · Authors · 2024-08-06
>
> We would first like to express our appreciation for your time and insightful comments. Please find our response to your concerns in the following:
>
> ---
>
> **1. The performance of InLine models and comparison with related works.**
>
> Thanks for the valuable comment.
>
> ***Firstly, under fair comparison, the improvement of our method is significant and consistent.***
>
> - We propose injective linear attention and local attention residual to address two core limitations of linear attention. The effectiveness of these two designs is validated in Table 3 and Table 4 of our paper. Table 3 shows that injective linear attention achieves significant ***accuracy gains of 2.5 and 9.8*** compared to vanilla linear attention. Table 4 shows that local attention residual further improves model performance ***from 80.2 to 82.4.*** We kindly argue that these improvements are significant and remarkable.
> - Our InLine attention module ***significantly improves performance*** across four representative architectures: ***DeiT, PVT, Swin and CSwin***. As shown in Table 5 and Table 6 of our paper, simply applying our InLine attention module to these four models leads to obvious accuracy gains. For example, InLine-Swin-S outperforms Swin-B with 57% of the Params and 56% of the FLOPs. These results demonstrate the superiority of InLine attention as an effective alternative to the widely used Softmax attention.
>
> ***Secondly, we offer additional comparison with more competitive related works like VVT.***
>
> - The primary focus of our paper is to demystify the limitations of linear attention, rather than achieving SOTA results. Therefore, we mainly focus on building our InLine models under the four simple baseline models. In contrast, VVT utilizes advance macro designs like ConvFFN to achieve more competitive results. For a fair comparison with VVT, we employ similar advance designs and provide the results below.
>
>   | Model  | #Params | FLOPs | Acc. |
>   | :----: | :-----: | :---: | :--: |
>   | VVT-L  |  61.8M  | 10.8G | 84.1 |
>   | InLine |  51.1M  | 6.8G  | 84.2 |
>
>   It is shown that ***our InLine model outperforms VVT with 20% fewer Params and 40% fewer FLOPs.***
>
> - Additionally, since our InLine model is extremely simple and effective, it delivers much better speed-accuracy trade-off than other competitive works. For example, as depicted in Fig. 1 of ***the PDF in general response***, our simple ***InLine-Swin model shows 1.8x-2.5x faster inference speed than VVT***, while achieving comparable accuracy. Speed is tested on a single RTX3090 GPU.
>
> - ***We will give more credits to these related works and include comparison in the revised manuscript.***
>
> ---
>
> **2. The analysis of local modeling capability.**
>
> Thanks for your insightful question. We offer clarification on the analysis of local modeling capability.
>
> - As shown in Eq. 6 of our paper, the local attention residual proposed in our paper is $\sum_{j=1}^9 r_j V_j^{N(i)}$, where $N(i)$ is the 3×3 neighborhood of $V_i$ and $V_j^{N(i)}$ represents the value token in this neighborhood, $j=1,\cdots,9$. Therefore, the attention local attention residual is not a simply residual $rV_i$ since it contains local property. ***The local property of the attention residual is beneficial for InLine attention, rather than the residual property.*** We offer additional experiments to further validate this.
>
> - ***Firstly, we replace the local residual with vanilla residual $rV_i$ and provide the results below.*** The window size is fixed as global, i.e. $56^2$.
>
>   |          Method           | Window | #Params | FLOPs | Acc. |
>   | :-----------------------: | :----: | :-----: | :---: | :--: |
>   |    InLine w/o residual    | $56^2$ |   30M   | 4.5G  | 80.2 |
>   | InLine + vanilla residual | $56^2$ |   30M   | 4.5G  | 80.2 |
>   |  InLine + local residual  | $56^2$ |   30M   | 4.5G  | 82.4 |
>
>   It can be seen that adding vanilla residual can not benefit InLine model and it achieves the same result as no residual. This indicates that the residual property is not the core of the proposed attention residual.
>
> - ***Secondly, we provide further ablation studies on $N(i)$.*** In our original design,  $N(i)$ is the 3×3 neighborhood of $V_i$. Here, we define $D_kN(i)$ as the dilated 3×3 neighborhood of $V_i$ with dilation $k$, where the concept "dilation" is the same as in dilated convolution. Thus we have $D_1N(i)=N(i)$. As $k$ enlarges, the local property of $\sum_{j=1}^9 r_j V_j^{D_kN(i)}$ is weakened.
>
>   |     Model     | $k$  | #Params | FLOPs | Acc. |
>   | :-----------: | :--: | :-----: | :---: | :--: |
>   | InLine-Swin-T |  1   |   30M   | 4.5G  | 82.4 |
>   | InLine-Swin-T |  2   |   30M   | 4.5G  | 81.7 |
>   | InLine-Swin-T |  3   |   30M   | 4.5G  | 80.8 |
>   | InLine-Swin-T |  4   |   30M   | 4.5G  | 80.6 |
>
>   As depicted in the above table, the model performance decreases significantly as $k$ enlarges, proving that local property is the real factor benefiting InLine model.
>
> - These results will be included in the revised version to make the analysis of local modeling capability more solid. Thanks again for your valuable question.

---

> > ### Comment · Reviewer_My8Q · 2024-08-12
> >
> > Thank you for the author’s rebuttal. It generally makes sense, and I will be raising my score accordingly. However, I want to emphasize that direct comparisons to closely related works (e.g., VVT) are essential. The explanations provided in the rebuttal are helpful in this regard.

---

> > > ### Author Response · Authors · 2024-08-13
> > >
> > > Thanks for your time and valuable comments. We will include these direct comparisons and discussions in our revised manuscript.

---

> ### Author Response · Authors · 2024-08-12
>
> Dear Reviewer My8Q, thank you for your insightful review and for engaging with our work. We would like to know if there are any additional questions or concerns. We are eager to engage in further discussion and provide clarification to fully address them.

---

### Official Review · Reviewer_TKLm · 2024-07-14

**Soundness:** 3
**Presentation:** 3
**Contribution:** 2
**Rating:** 5
**Confidence:** 4

**Summary:**

While linear attention reduces the quadratic complexity of softmax attention, it often suffers from inferior performance. The authors analysed the reason behind it and identified two crucial properties which linear attention lacks: 1) injectivity where different queries in linear attention may have the same attention scores, increasing semantic confusion; 2) local modeling where linear attention can’t capture local patterns well. To bridge the gap, the authors proposed injective linear attention (InLine) with local enhancement, which achieves comparable and even better performance than softmax attention across several models and benchmarks.

**Strengths:**

1) The analysis of injectivity and locality includes both theoretical understanding and empirical evidence;
2) InLine achieves competitive performance to softmax attention, and performs better than several previous linear models.

**Weaknesses:**

1) The locality issue of linear attention has been discussed in-depth before;
2) The motivation and formulation of InLine are very similar to FLatten, and doesn’t show substantial quality gain to FLatten;
3) It would be great to have language modeling experiments;

**Questions:**

1) This is not the first paper discussing the locality problem of linear attention. Check [1] for more details.
2) The intuition of InLine is very similar to FLatten. For example, FLatten removes the division operation so it satisfies the injectivity and Flatten adopts depthwise convolution, enhancing locality modeling. Based on Table 9, InLine doesn’t outperform FlAtten significantly. The authors should give a more comprehensive analysis on the similarity and differences compared to Flatten and why InLine is preferred.

[1] Qin et al., 2022; The Devil in Linear Transformer.

**Limitations:**

The authors discussed the limitations of their work.

---

> ### Author Rebuttal · Authors · 2024-08-06
>
> We would first like to express our appreciation for your time and insightful comments. Please find our response to your concerns in the following:
>
> ---
>
> **1. The locality problem and related work.**
>
> Thanks for pointing out this important related work [1]. Here, we offer clarification on the relationship between this study and our work.
>
> ***Firstly**, our work verifies the vital importance of the injective property of attention mechanism.* This unique and valuable perspective helps us develop the extremely simple yet effective injective linear attention.
>
> ***Secondly**, while our work shares some similar analyses on the locality problem with [1], it has fundamental distinctions:*
>
> - ***We thoroughly validate the crucial role of local modeling in attention mechanism, while [1] mainly focuses on identifying differences.*** There could be many different behaviors between Softmax and linear attention, but a large number of them may not be the key disparities. Therefore, our study conducts a series of experiments in Fig. 3, 4 and Table. 2, 4 to fully validate that: (1) Softmax attention's effectiveness depends significantly on robust local modeling. (2) Varied local priors contribute substantially to the performance gap between Softmax and linear attention. In contrast, [1] observes that Softmax attention is more concentrated locally than linear attention (see its Fig. 2), but it does not verify whether this is a key factor leading to the performance gap between the two.
>
> - ***The design proposed in our paper is more effective than [1].*** To enhance locality, [1] proposes DiagAttention, which restricts each token's receptive field to its local block. This approach resembles the window attention in Table 4 (left), which shows little improvement (window=$14^2$, acc=80.4) compared to vanilla global linear attention (acc=80.2). In contrast, our design boosts accuracy significantly from 80.2 to 82.4, fully unleashing the power of global linear attention. Additionally, we offer a comparison between our InLine model and TransNormer proposed in [1] under Swin-T structure. Our InLine model achieves better results.
>
>   |   Method    | #Params | FLOPs | Acc. |
>   | :---------: | :-----: | :---: | :--: |
>   | TransNormer |   30M   | 4.5G  | 79.9 |
>   |   InLine    |   30M   | 4.5G  | 82.4 |
>
> - Thanks for pointing this out again. ***We will include discussion and give more credits to this work in the revised manuscript.***
>
> [1] Qin et al., 2022; The Devil in Linear Transformer.
>
> ---
>
> **2. The similarities and differences compared to FLatten.**
>
> Thanks for your valuable comment.
>
> ***Firstly**, removing the division operation can make FLatten injective but hinders performance.* Without the division operation, FLatten can not ensure the attention weights sum up to 1, which is important for stabilizing the model. Therefore, when the division operation is removed from FLatten-Swin-T, the model experiences an accuracy drop of 0.2.
>
> ***Secondly**, we offer a detailed analysis of the similarities and differences with FLatten.*
>
> - ***Our work and FLatten are largely orthogonal.*** Our work analyzes the injectivity and locality of Softmax and linear attention, while FLatten discusses the focus ability and feature diversity of linear attention. Furthermore, we present subtraction normalization to make linear attention injective,
>   $$
>   \rm{InL_K}(Q_i)=\left[\phi(Q_i)^\top \phi(K_1), \cdots, \phi(Q_i)^\top \phi(K_N)\right]^\top-\frac{1}{N} \sum_{s=1}^N \phi(Q_i)^\top \phi(K_s)+\frac{1}{N},
>   $$
>   while FLatten introduces a specific mapping function $\phi=\frac{||x||}{||x^{**p}||}x^{**p}$ to sharpen attention distribution. These methods are orthogonal and can be employed together.
>
> - ***The findings of our paper are more fundamental***, and could be more beneficial for the community in designing effective linear attention patterns. As discussed in our paper (L145-L152), the focus ability identified in FLatten can be viewed as a special case of non-injectivity and confusion problem. And our studies on locality can also explain why dwconv improves FLatten.
>
> - ***Our InLine attention is more effective and efficient than FLatten.*** The proposed InLine attention is extremely simple yet effective, yielding both faster speed and higher accuracy compared to FLatten. For example, a single InLine attention block is 1.8x faster than a FLatten block, and InLine-PVT-T achieves 0.4 accuracy gain with 1.2x speed up than FLatten-PVT-T. These results are tested on a single RTX3090 GPU.
>
> ---
>
> **3. Language modeling experiments.**
>
> Please refer to our general response.

---

> ### Author Response · Authors · 2024-08-12
>
> Dear Reviewer TKLm, thank you for your insightful review and for engaging with our work. We would like to know if there are any additional questions or concerns. We are eager to engage in further discussion and provide clarification to fully address them.

---

### Author Rebuttal · Authors · 2024-08-06

We thank all the reviewers for their insightful and valuable comments.

We have carefully considered the reviewers' comments and provided additional clarification to address each concern. Here, we offer general responses to all reviewers on two key issues.

---

**1. Discussion with related works.**

We appreciate the reviewers for highlighting several important related works that we have overlooked. Detailed discussions with these studies are provided in separate responses to each reviewer. We will give more credits to these works and include the discussions in the revised manuscript.

---

**2. Applying InLine attention to language models.**

- The proposed injective linear attention can be applied to language models. Currently, our work mainly focuses on vision tasks since we follow the line of linear attention studies in vision [1,2,3,4]. However, we fully agree with the reviewers' comments that applying our method to language models can greatly improve its impact. Here, we provide detailed discussion on how to apply injective linear attention to language models.

- When applied to auto-regressive language models, our injective linear attention can naturally achieve parallel training and $\mathcal{O}(1)$ complexity (per token) inference.

- ***Parallel training.***

  The Eq. 4 of our paper can be written in a two-step form as follows:
  $$
  A=\phi(Q)\phi(K)^\top,\ \ \mathrm{InL}=A-(A\cdot1)\odot l+l,
  $$
  where $Q,K\in\mathbb{R}^{N\times d}$ are query and key, $A\in\mathbb{R}^{N\times N}$ represents the raw attention scores, and $1,l\in\mathbb{R}^{N\times 1}$ are vectors with all 1 elements and all $1/N$ elements, respectively. We could see that $(A\cdot1)\odot l\in\mathbb{R}^{N\times 1}$ denotes the row mean of $A$ and the above formulation is equivalent to Eq. 4. To apply InLine attention to language models, the only modification we need to make is applying the causal mask. To achieve this, the above equation is rewritten as:

  $$
  A=\phi(Q)\phi(K)^\top,\ \ \mathrm{InL}=(A-((A\odot M)\cdot1)\odot l+l)\odot M,
  $$
  where $M\in\mathbb{R}^{N\times N}$ is the casual mask, and $l\in\mathbb{R}^{N\times 1}$ is a vector whose $i$-th element is $1/i$. Similarly, $((A\odot M)\cdot1)\odot l\in\mathbb{R}^{N\times 1}$ is the row mean of the causal attention matrix $A\odot M$. In this way, the InLine attention map with causal relationship is obtained, and the output is $O=\mathrm{InL}\cdot V$.

- ***Recurrent inference.***

  The causal InLine attention can be written in a recurrent mode, supporting $\mathcal{O}(1)$ complexity inference. This recurrent form is formulated as follows:
  $$
  S_i=S_{i-1}+\phi(K_i)V_i^\top,\ \ W_i=W_{i-1}+\phi(K_i),\ \ Z_i=Z_{i-1}+V_i,\ \ O_i=\phi(Q_i)^\top S_i-\phi(Q_i)^\top W_i/i+Z_i/i
  $$

  where $Q_i,K_i,V_i\in\mathbb{R}^{d}$. This recurrent form is ***strictly equivalent to*** the parallel form.

- The above analyses show that our injective linear attention can apply to auto-regressive language models, enjoying both parallel training and $\mathcal{O}(1)$ complexity (per token) inference. In addition, chunk-wise parallel training like [5] can also be used, which is a mix of the parallel and recurrent modes.

- Due to time and computation resource constraints, we are still working on building InLine language model and are unable to offer the results here. Hopefully, we could possibly provide the results in the revised manuscript.

[1] Hydra attention: Efficient attention with many heads. In ECCVW, 2022.

[2] Efficient attention: Attention with linear complexities. In WACV, 2021.

[3] Soft: Softmax-free transformer with linear complexity. In NeurIPS, 2021.

[4] Flatten transformer: Vision transformer using focused linear attention. In ICCV, 2023.

[5] Retentive network: A successor to transformer for large language models. ArXiv, 2307.08621.

---

**For detailed responses to individual reviewer comments, please refer to our separate responses to each reviewer.**

Lastly, we would like to thank the reviewers for their time and we are welcome for any further discussion.

---

### Decision · Program_Chairs · 2024-09-25

**Decision:**

Accept (poster)

**Comment:**

The paper received 5/5/5/6 ratings. Reviewers had initial concerns regarding the novelty of the paper, the similarity between the paper and related works, missing comparison with Mamba and other related works, lack of ablation study, etc. The authors provided a rebuttal which successfully addressed those concerns. After checking paper, reviews and rebuttal, AC recommends to accept the paper.